# Inferring Dynamic Physical Properties from Video Foundation Models

## Abstract

We study the task of predicting dynamic physical properties from videos. More specifically, we consider physical properties that require *temporal* information to be inferred: elasticity of a bouncing object, viscosity of a flowing liquid, and dynamic friction of an object sliding on a surface. To this end, we make the following contributions: (i) We collect a new video dataset for each physical property, consisting of synthetic training and testing splits, as well as a real split for real world evaluation. (ii) We explore three ways to infer the physical property from videos: (a) an oracle method where we supply the visual cues that intrinsically reflect the property using classical computer vision techniques; (b) a simple read out mechanism using a visual prompt and trainable prompt vector for cross-attention on pre-trained video generative and self-supervised models; and (c) prompt strategies for Multi-modal Large Language Models (MLLMs). (iii) We show that video foundation models trained in a generative or self-supervised manner achieve a similar performance, though behind that of the oracle, and MLLMs are currently inferior to the other models, though their performance can be improved through suitable prompting.

## 1 Introduction

Humans are remarkably adept at intuitively estimating physical properties from visual observations. Without direct interaction, people can often estimate how bouncy a ball is, how thick a liquid seems, or how slippery a surface might be—simply by watching how objects move. While these estimations are not precise in a scientific sense, they are sufficiently accurate for guiding perception, prediction, and action. Bringing this capability to machines is an important step towards building more general and physically grounded artificial intelligence. In particular, visual systems that can infer dynamic physical properties from raw video could enhance robotic manipulation, embodied agents, and video understanding tasks in ways that go beyond the traditional perception tasks of recognition, detection, and segmentation.

Recent progress in video foundation models, including generative models (Xing et al., 2024; Liu et al., 2024b), self-supervised models (Bardes et al., 2023; Assran et al., 2025) and multi-modal large language models (MLLMs) (Hui et al., 2024; Comanici et al., 2025; Hurst et al., 2024), have shown impressive capability in synthesizing realistic dynamics, learning general-purpose video representations, and tackling semantic understanding tasks, for example, video question answering. However, a question that remains underexplored is: **do these models acquire an understanding of *dynamic* physical properties from videos ?**

In this paper, we address this question by focusing on several representative physical properties that are not directly observable in static frames but instead emerge through temporal dynamics: the elasticity of a bouncing object, the viscosity of a flowing liquid, and the dynamic friction between a surface and a sliding object. These properties are especially compelling because their inference requires temporal reasoning and sensitivity to subtle visual cues—such as deformation, deceleration, spreading, or oscillation. By examining how well current video foundation models capture these dynamic attributes, we aim to assess their physical understanding beyond static appearance.

To support this investigation, we introduce a new dataset, *PhysVid*, specifically designed to evaluate the dynamic physical properties from video. Existing datasets lack ground-truth annotations for such properties, so we construct *PhysVid* using a combination of synthetic videos—rendered via a physics

simulator—and real-world videos sourced from the internet or captured in-house. Each video is annotated with physical property values, either derived from simulation parameters or estimated manually. The dataset is designed to facilitate the study of out-of-domain generalization, both within the synthetic domain and from synthetic to real-world data. To establish an upper bound on what is inferable from visual input alone, we implement an oracle method for each property. These oracles leverage privileged access to the visual cues that directly reflect the corresponding property.

We evaluate three categories of video foundation models: generative models, self-supervised models, and multi-modal large language models (MLLMs). For the generative and self-supervised models, we propose a simple yet effective readout mechanism that extracts dynamic physical properties from pre-trained, frozen representations. Our method introduces a learnable query vector that attends to internal representation tokens via cross-attention, enabling the selective extraction of relevant information. This approach is both lightweight and training-efficient. For MLLMs, we explore various prompting strategies to elicit predictions of dynamic physical properties directly from video input. These strategies include few-shot prompting to provide task context, as well as procedural prompting that guides the model through the oracle estimation steps—helping it focus on the intrinsic visual cues that reveal the target properties.

## 2 RELATED WORK

**Physics Prediction from Images and Videos.** Inferring physical properties from visual observations remains a core challenge in computer vision. Early methods estimate latent physical parameters (e.g., mass, friction, stiffness) via differentiable physics engines or learning-based simulators (Wu et al., 2015; Ding et al., 2021; Jatavallabhula et al., 2021; Li et al., 2020; Wang et al., 2020a; 2018), while later works infer salient attributes like viscosity or elasticity from task-specific visual cues (Kawabe et al., 2014; Paulun et al., 2015; Assen et al., 2018; Norman et al., 2007; Kawabe & Nishida, 2016; Paulun et al., 2017; Paulun & Fleming, 2020), yet both rely heavily on simulation supervision, domain priors, or handcrafted heuristics. More recently, unsupervised learning of intuitive physics has emerged via next-frame prediction from large-scale everyday physical scenes (Voleti et al., 2022; Lu et al., 2023; Agrawal et al., 2016; Finn & Levine, 2017; Babaeizadeh et al., 2021; Hafner et al., 2019; Fragkiadaki et al., 2016; Garcia et al., 2025), capturing latent dynamics without explicit physical supervision. However, the resulting representations are usually implicit and lack interpretability in terms of concrete physical quantities. In contrast, we infer physical properties by directly prompting pre-trained video foundation models, enabling explicit estimation without reliance on task-specific heuristics, or end-to-end prediction pipelines from scratch.

**Physics Datasets and Benchmarks.** An increasing number of physics-related datasets have been collected in recent years to provide ground truth annotations for different physical properties, including material (Sharma et al., 2023; Gao et al., 2024), shadow (Wang et al., 2020b; 2021), support relations (Silberman et al., 2012),occlusion (Zhan et al., 2022; 2024a), mass and volume (Wu et al., 2016). Another line of work (Chow et al., 2025; Shen et al., 2025; Riochet et al., 2018; Bordes et al., 2025; Tung et al., 2023; Bear et al., 2021) proposes broad benchmarks with video-image-text QA tasks to assess physical understanding in vision-language models, but the questions are typically qualitative and categorical. In contrast, our datasets consist of both *synthetic* and *real-world* videos annotated with the *quantitative value* for the associated physical parameter of the coefficient of friction, elasticity, and viscosity.

## 3 PROBLEM SCENARIO AND THE *PhysVid* DATASETS

In this paper, we address the problem of estimating physical properties from videos. Specifically, we focus on three properties: **elasticity** of a bouncing object, **viscosity** of a flowing liquid, and the **dynamic friction coefficient** between a surface and a sliding object. Given a video $v \in \mathbb{R}^{T \times H \times W \times 3}$, we consider two formulations, the first is **absolute value prediction**, where the input is a single video and the model is tasked with predicting the numerical value of the physical property, *i.e.*, $y_{\text{abs}} = \Phi(v; \theta_1)$. The second is **relative value comparison**, where the input is a pair of videos captured from the same viewpoint, and the model must determine whether the first video exhibits a higher physical property value than the second, *i.e.*, $y_{\text{rel}} = \Phi(v_1, v_2; \theta_2)$, and $y_{\text{rel}}$ is binary.

Figure 1: **Examples of the *PhysVid* dataset.** Each row shows a different property, and each column shows three frames from video samples in the synthetic sets (`train`, `test-1`, and `test-2`) and the real `test-3` set. The `train` and `test-1` sets are from the same distribution. In `test-2` parameters, such as lighting, viewpoint and color, differ from those in `test-1`.

Each scenario is parameterized by a set of variables, including the value of the target *physical property* (*e.g.,* elasticity, viscosity, or friction), and a set of *nuisance parameters* (including camera viewpoint, object appearance, lighting, *etc.*). While the model must be sensitive to changes in the physical property, it should be robust (ideally invariant) to variations in nuisance parameters.

To assess generalization, we define two domains of nuisance parameters, denoted as $\mathcal{A}_1$ and $\mathcal{A}_2$, which differ in their distributions. For instance, $\mathcal{A}_2$ may have different camera viewpoints or different lighting conditions to $\mathcal{A}_1$ (full details of these differences are given in Supplementary Section B). We generate a dataset using a physics-based simulator, consisting of one training split and two test splits. The models are only trained on the training split from the simulator for all the evaluations. The training and `test-1` splits are sampled from $\mathcal{A}_1$, sharing the same nuisance distribution; `test-2` is drawn from $\mathcal{A}_2$, introducing a distribution shift. The target property values are sampled from a shared range across all splits to ensure consistency. Finally, `test-3` consists of real-world videos, used to evaluate generalization beyond simulation.

### 3.1 THE *PhysVid* DATASETS

To study the dynamic physical properties of elasticity, viscosity, and friction, we construct a dataset for each, containing both synthetic and real-world videos. Synthetic ones are generated with the Genesis simulator (Zhou et al., 2024), and real ones are captured with an iPhone in slow-motion mode or downloaded from the Internet. For each property we have: $10,000$ videos for `train`; $1000$ videos for each of `test-1` and `test-2`; and $100$ videos for `test-3`. Sample frames are shown in Figure 1, and sample videos are attached as supplementary. In the following we describe how each property is realized in the video. Please refer to Supplementary Section B for more details of the datasets.

**Elasticity**

We study an object's elasticity by analyzing the motion of a ball dropped onto the ground and its subsequent bounces. In physics, elasticity $e$ is quantified as the ratio of the rebound velocity $v_{\text{after impact}}$ to the impact velocity $v_{\text{before impact}}$, and also equals $\sqrt{h_{\text{bounce}}/h_{\text{drop}}}$, where $h_{\text{drop}}$ is the dropping height and $h_{\text{bounce}}$ is the bouncing height. Here and for the following properties, please refer to Supplementary Section C for the detailed derivations. These expressions are used for the oracle estimation in Section 4.1.

*Synthetic Dataset.* All synthetic videos are generated using Genesis (Zhou et al., 2024), with object's elasticity as the target property. Nuisance factors include drop height, camera viewpoint, object appearance, and lighting conditions. The object is of the same size in all videos. Note, here and for the following properties, the ground truth property value is obtained directly from the simulator.

*Real-World Dataset.* The real-world videos are collected from YouTube using the search term "ball bouncing experiments". Each clip is manually trimmed to include the drop-and-bounce sequence of a single ball. The dataset includes a wide range of materials (*e.g.*, rubber balls, tennis balls, basketballs, balloons, *etc*), resulting in diverse elasticity values. The ground truth elasticity values for the real sequences are estimated by computing $\sqrt{h_{\text{bounce}}/h_{\text{drop}}}$: the videos are chosen such that the balls bounce in a fronto-parallel plane, which means that ratios of image heights (differences in $y$-coordinates) are approximately equal to the ratio of heights in 3D. These image differences are obtained by manual annotation.

**Viscosity**

We study the viscosity by observing a liquid column dropping and spreading on the ground. The viscosity can be reflected by the growth rate of the liquid area on the ground. The viscosity $\mu$ is negatively correlated to the liquid area growth rate $\frac{d(A(t))}{dt}$, given the controlled liquid density $\rho$, controlled liquid column diameter $D$, and controlled dropping velocity $v$ of the liquid column when it reaches the ground.

***Synthetic Dataset.*** The synthetic videos are generated using Genesis (Zhou et al., 2024), where the target property is the viscosity of liquid. Nuisance factors include camera viewpoint, object appearance, and lighting conditions. The liquid column is of the same size in all videos.

***Real-World Dataset.*** Since it is challenging to find real-world videos online that provide ground-truth viscosity values while controlling for other relevant physical parameters—such as $\rho$, $D$ and $v$, we collected real videos under controlled conditions. We use a funnel with a fixed nozzle diameter to produce a consistent liquid column. A funnel holder allows us to fix the height from which the liquid is poured, thereby controlling the initial velocity $v$. Ground-truth viscosity values for each liquid are obtained from standard physics reference tables. The selected liquids span a wide range of viscosities, from 1.2 (*e.g.*, coffee) to 225 (*e.g.*, maple syrup), allowing for a diverse and comprehensive evaluation.

### Friction

We study friction between an object and a surface by observing how the object slows down as it slides with an initial velocity. The dynamic friction coefficient $\mu_k$ is proportional to the (negative) acceleration of the object $a$.

***Synthetic Dataset.*** The synthetic videos are generated using Genesis (Zhou et al., 2024), where the target property is the dynamic friction coefficient at the contacting surface of the object and the ground. Nuisance factors include initial location and initial velocity of the object, camera viewpoint, object appearance, and lighting conditions. The object is of the same size in all videos.

***Real-World Dataset.*** While many online videos depict objects sliding on surfaces, they lack ground-truth annotations for friction coefficients. We therefore collect a real video dataset featuring 5 different objects and 6 surface materials, spanning a wide range of dynamic friction values. Each object is given an initial velocity by sliding it down from a slope and it then slides on a horizontal plane. To obtain ground-truth friction coefficients, we use a spring dynamometer to measure the friction force $F$ for each object-surface pair (by dragging the object at constant speed), and record the object's weight $G$. The dynamic friction coefficient is then computed as: $\mu_k = F/G$.

## 4 INFERRING PHYSICAL PROPERTIES

This section presents the three different ways for inferring dynamic physical properties: an oracle method via classical computer vision techniques (Section 4.1); a visual prompt mechanism for video generative and self-supervised models (Section 4.2); and prompts for MLLMs (Section 4.3).

### 4.1 ORACLE ESTIMATION

**Elasticity.** We aim to estimate elasticity from both synthetic and real-world videos. The key visual cue is the relative height of the ball during its drop and subsequent bounce, observed in 3D. As noted earlier, the ratio in 3D can be approximated from their corresponding image-space measurements. This approximation is exact when the motion occurs in a fronto-parallel plane, and remains reasonably accurate otherwise—since the ratio of lengths between parallel line segments is invariant under affine transformations (Hartley & Zisserman, 2004). Given that perspective effects are minimal in our videos, the affine approximation provides a reliable estimate for elasticity. To automate this process, we extract the ball's trajectory $y(t)$ from the video and input the sequence of ratios into a GRU network to regress the elasticity. In detail, we segment the ball in each frame and use their centroids as the $y$-coordinate. From this trajectory, we identify key points: the initial drop position, the first ground contact, and the peak of the first bounce. The resulting trajectory is normalized to the range $[0, 1]$, by subtracting the $y$-coordinate of the first ground contact and dividing by the initial drop height. This normalization not only ensures invariance to viewpoint and scale, but also simplifies learning for the GRU by standardizing the input distribution. We train a GRU, as it is noisy to directly obtain $h_{\text{drop}}$ and $h_{\text{bounce}}$ using heuristics (*e.g.,* determining the maximum and minimum

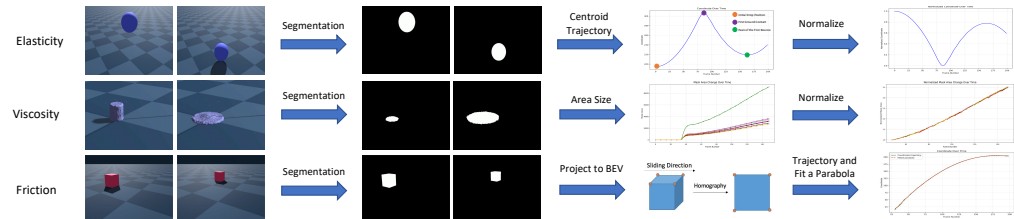

Figure 2: **Oracle methods for physical properties.** The objective in each case is to extract a measurement from the sequence that can directly be used to predict the property. For elasticity, we extract the centroid trajectory from segmentation masks, and then normalize the $y$-coordinates into 0-1; the ratio of bouncing to dropping height over the sequence indicates the elasticity. For viscosity, we calculate the area size in the image via segmentation masks, and then normalize the area sizes by the area in the frame when the liquid first touches the ground; the slope of the normalized area size sequence reflects the viscosity. For friction, we transform to a bird's eye view (using a homography transformation based on 4 corner points of the top surface of the sliding object), and fit a parabola $x = \alpha t^2 + \beta t + c$ to the transformed trajectory; the parabola coefficient $\alpha$ predicts the friction coefficient. For each video, we show the segmentation for two frames (left $\rightarrow$ right).

points), and in practice a GRU provides a good estimate. The full pipeline is illustrated in Figure 2 (top row). For the **absolute prediction**, the normalized trajectory is fed into a GRU network, which directly regresses the elasticity value. For the **relative comparison**, the binary decision score between two videos $v_1$ and $v_2$ is calculated as:

$$\texttt{score} = \sigma(\log(\frac{e_1}{e_2})), \tag{1}$$

where $e_1$ and $e_2$ are the estimated elasticities based on height ratios, and $\sigma(\cdot)$ denotes the sigmoid function.

**Viscosity.** The key visual cue for estimating viscosity is the rate at which the liquid spreads on the ground-plane, measured as an area ratio normalized by the initial area of the liquid column. As with elasticity, we approximate perspective using an affine transformation – here of the ground-plane. Since area ratios are invariant under affine transformations (Hartley & Zisserman, 2004), the liquid's normalized image-space area growth approximates its true normalized ground-plane expansion (in our setup the liquid spreads only within a limited area around the release point, and the camera is distant; consequently an affine viewing approximation is adequate). Specifically, we extract segmentation masks for each frame and compute the liquid's area over time. This area sequence is normalized by the area in the first frame where the liquid contacts the surface, ensuring invariance to viewpoint and scale. The process is illustrated in Figure 2 (middle row). For **absolute prediction**, we calculate the slope $k$ of $A(t)$ and use $1/k$ to represent the viscosity value; For **relative comparison**, the binary decision score between two videos $v_1$ and $v_2$ is calculated as in Equation 1, where $e_1$ and $e_2$ are the estimated viscosities based on area growth rate.

**Friction.** The key visual cue for estimating dynamic friction is the acceleration of the sliding object—*i.e.*, how quickly its velocity decreases due to friction—which can be inferred from its position over time. Since the object moves significantly in the video, we do not use an affine approximation, but instead take account of the projective geometry by mapping the object's motion to a bird's-eye view, allowing for consistent trajectory analysis. This is achieved by estimating a homography between the image and bird's eye view (normal to the plane) from the four corners of the object's top surface (see Figure 2, bottom row). We fit a parabola $x = \alpha t^2 + \beta t + c$ to the transformed top surface trajectory to estimate the acceleration $a$ from the coefficient $\alpha$, and the coefficient of friction $\mu_k = 2\alpha/g$. For **absolute prediction**, we use the estimated $\mu_k$ to represent the friction coefficient value; For **relative comparison**, the binary decision score between two videos $v_1$ and $v_2$ is calculated as in Equation 1, where $e_1$ and $e_2$ are the estimated friction coefficients based on the transformed object trajectory.

## 4.2 VIDEO GENERATIVE AND SELF-SUPERVISED MODELS

**Video Feature Extraction**

Figure 3: **Architectures for dynamic physical property prediction. Left**: video generative model as backbone; **Middle**: video self-supervised model as backbone; **Right**: multimodal large language model (MLLM). For the pre-trained video diffusion model (U-Net, left) and the pre-trained self-supervised model (ViT, middle), the representations are kept frozen, and a 'visual prompt' learns to infer the physical properties. For the MLLMs, the physical properties are inferred using a language prompt (right).

Given a video $v \in \mathbb{R}^{T \times H \times W \times 3}$, we extract features with a pre-trained video backbone, that can either be generative or self-supervised, resulting into spatiotemporal feature representations, *i.e.*, $r = \psi(v) \in \mathbb{R}^{t \times h \times w \times c}$, which can be detailed as follows.

***Generative Model as Backbone.*** We adopt a pre-trained video diffusion model (Figure 3, left), namely DynamiCrafter (Xing et al., 2024), to compute the visual features. Specifically, given an input video, we add noise to the latent representations after the pre-trained VAE encoder, and replace the text prompt with a learnable embedding. We extract multi-scale features from all U-Net layers at diffusion time step 50, which was shown to be effective for capturing 3D physics in prior work (Tang et al., 2023; Zhan et al., 2024b). To aggregate the features, we introduce a learnable query vector $q$, which is first mapped to the different dimensions of the multi-scale features (see Supplementary Section A.2 for details), and then attends to the diffusion tokens ($r_i$) via cross-attention: $p = \sum_{i=1}^{t \times h \times w} \mathtt{softmax}(q \cdot r_i) \cdot r_i$. The resulting vectors $p$ from different layers are then mapped by another MLP network to a common dimension and average pooled to generate the final video feature representation $P$. To predict the physical properties, we train the text token of the generative model, together with the 'visual prompt' architecture that includes the query $q$ and the MLPs.

***Self-Supervised Model as Backbone.*** Here, we adopt a pre-trained self-supervised model (Figure 3, middle), namely V-JEPA-2 (Assran et al., 2025), as the visual backbone. The input video is passed through the model, and we extract feature tokens from all layers of the ViT encoder. Similar to the generative setting, we introduce a learnable query vector $q$ to extract the video feature representation $P$ from the ViT tokens via attentive pooling. Although the feature dimension at each ViT layer is the same, we still use a MLP network to map $q$ to generate the query vector of each layer (keeping it similar to the generative setting in terms of MLP network architecture), and use another MLP network to map the output vectors $p$ to a same dimension as the generative setting before average pooling them to get $P$. Please see Supplementary Section A.2 for more details.

**Physical Property Prediction**

Given the computed feature $P$ from video foundation models, we train a MLP network to predict the physical properties using the synthetic video dataset training split. The network for each property is trained separately.

***Absolute Value Prediction.*** Given the resulting video feature ($P$), we pass it through a MLP network $\gamma$ to predict the absolute value $\chi$ of the physical property: $\chi = \gamma(P)$. For elasticity and friction, the absolute value prediction is supervised with L1 loss with the ground truth value; For viscosity, as the ground truth values may have very different scales, *i.e.,* from $1e-5$ to $1e-2$, the absolute value prediction is trained with Log L1 loss, which calculates L1 loss between the log of the predicted value and the log of the ground truth value.

***Relative Value Prediction.*** Given the resulting features for a pair of videos, $P_1$ and $P_2$, we concatenate them and formulate a binary classification problem, indicating which video has a larger physical property value via a MLP network $\gamma$: $\xi = \gamma([P_1, P_2])$. The binary prediction for all three tasks is trained with binary cross entropy loss with the binary ground truth.

***Bridging the Sim2real Gap.*** Since our models are trained on synthetic datasets, they may not generalize well to real-world test videos due to the domain gap. To mitigate this sim-to-real gap, for both synthetic training and real test, we draw a red circle on each video frame, enclosing the full trajectory of the target object or liquid, as illustrated in Figure 4 (middle). The red circle is ob-

tained automatically as a bounding ellipse enclosing the merged masks of the target object or liquid across all frames. This visual cue directs the model's attention to the relevant region (Shtedritski et al., 2023), effectively signaling which object to focus on for physical reasoning. The red circle serves as a lightweight yet effective form of weak annotation that helps the model localize and interpret the dynamics of interest. Please refer to Supplementary Section G for the quantitative results demonstrating the effectiveness of drawing such red circles to mitigate the sim-to-real gap.

### 4.3 MULTIMODAL LARGE LANGUAGE MODELS

This section studies off-the-shelf multimodal large language models (MLLMs) for understanding dynamic physical properties from video. We explore various prompting strategies on state-of-the-art MLLMs, including Qwen2.5-VL-Max (Hui et al., 2024), GPT-4o (Hurst et al., 2024), and Gemini 2.5 Pro (Comanici et al., 2025), as illustrated in Figure 3 (right). Examples of the prompting strategies are provided in Supplementary Section E.

**Preliminary.** The MLLM receives video frames as visual input. The text prompt includes (1) a brief description of the target property—for example: "we are studying the viscosity of the liquid, where water is 1.0 and honey is 5000.0." This is followed by (2) a query, such as: "what is the viscosity value of the liquid in the video?" (absolute) or "which video shows a liquid with higher viscosity? please output a decision score between 0 and 1, indicating the likelihood that the first video exhibits a higher property value." (relative). All the following prompt strategies provide (1) and (2) by default, and we note the differences and extensions.

**Baseline Prompt.** For *relative* tasks, we specify that the first $n$ frames belong to the first video and the last $n$ to the second.

**Black Frames in Between.** For the *relative* setting, we insert black frames between the two video segments to clearly separate them. In the prompt, we refer to the videos as the frames before and after the black frames, rather than as the first and last $n$ frames.

**Few-Shot Examples.** For both *relative* and *absolute* settings, we provide several examples, including the video input and desired ground truth. For fair comparison with visual prompting, we use examples in the synthetic training split.

**Frame Index Provided.** For both *relative* and *absolute* settings, we input the text of the index of each frame along with the frames. In this way the MLLMs may have a better understanding about the temporal relations between the input video frames.

**Oracle Estimation Teaching.** For both *relative* and *absolute* settings, we provide the key cue to concentrate on from the *PhysVid* Datasets section description to teach the MLLM how to estimate the properties step by step.

## 5 EXPERIMENTS

**Implementation Details.** During oracle estimation, we train the GRU network with a learning rate of $1e-3$ and the batch size is 128. For the generative and self-supervised video models, the backbones are frozen, the trainable parameters are optimised with a learning rate of $1e-5$ and the batch size 16. For MLLMs, we perform promp selection, and use the best strategy that we find for each of the absolute and relative settings for the experiments. *Few-shot examples* and *oracle estimation teaching* work best for the absolute and relative settings, respectively, as they directly provide the model with more context information about the properties. Please refer to Supplementary Section D for the comparison results and analysis. All models are trained on H100/A6000/A40 GPUs. Please refer to Supplementary Section A for more implementation details.

**Evaluation Metrics.** For *relative value comparison*, we report the ROC AUC score; for *absolute value prediction*, we use the Pearson Correlation Coefficient between the prediction and ground truth as this automatically calibrates the predictions to the scale of the ground truth. Please refer to Supplementary Section A.4 for more details and motivations on the evaluation metrics.

Figure 4: **Qualitative results. Top Left**: An example for elasticity absolute value prediction; **Bottom Left**: An example for friction relative value comparison. For each example, the original input video is shown on the left. A static red circle is overlaid in the center to highlight the full trajectory of the object on every frame, shown in the middle. Model predictions are shown on the right, including results from the Video Generative Model (VGM), Video Self-Supervised Model (VSM), and a MLLM (Gemini in this case). For the relative formulation, the ground truth value of '1' indicates that the first (top) video has larger dynamic friction coefficient than the second video. In this example, the initial velocity of the lego brick in the two videos is similar (note the same displacement from frame 0 to 2), but the velocity reduces to 0 at frame 30 in the first video, while the object is still moving in frame 30 to 60 in the second video. **Right**: Scatter plots of prediction vs ground truth for the elasticity property from the V-JEPA-2 model.

## 5.1 RESULTS FOR RELATIVE VALUE COMPARISON

Table 1 (left) shows relative value comparison results across physical properties and model types. The oracle estimator performs nearly perfectly on `test-1` and `test-2`, and strongly on `test-3`, indicating that the task is largely solvable using visual cues, geometry, and physics. Both generative and self-supervised video models achieve strong results on synthetic splits (`test-1` and `test-2`). Notably, they can also generalize well to the real-world split (`test-3`) for viscosity and elasticity, which rely on simple height ratios and expansion. However, friction proves more challenging. Models trained on synthetic data struggle to generalize, likely due to the fact that reliance on visual references (*e.g.*, ground plane grids) is absent in real videos, and due to friction's inherent complexity involving higher-order motion and projective geometry of the viewpoint. To further confirm, we introduce an additional real-world training split for friction videos with disjoint objects and surfaces from the test set (see Supplementary Section B.2 for more details). Fine-tuning the visual prompting architecture on this data improves performance on the real test split, as shown by the * values in Table 1. Multimodal large language models (MLLMs), though not working very well with *Baseline Prompt* (see Supplementary Section D), when prompted properly, also perform well, especially on real videos, which are more *in-distribution* for them – while on synthetic splits, their performance drops significantly. This is likely due to the fact that the models tend to leverage semantic cues rather than visual motion.

## 5.2 RESULTS FOR ABSOLUTE VALUE PREDICTION

Table 1 (right) shows results for absolute value prediction across physical properties and methods. This task is more challenging than relative comparison, as models must regress quantitative physical values rather than compare video pairs from the same viewpoint. Similar to the relative setting, the oracle estimator achieves near-perfect performance on `test-1` and `test-2`, and strong performance on `test-3`, confirming that the task is largely solvable through visual cues, multi-view geometry, and physical laws. We highlight several key observations: (i) **comparable performance across backbones**. Despite being trained for generative tasks, video generative models perform on par with self-supervised models when predicting dynamic physical properties. (ii) **friction remains challenging**. Similar to the relative setting, both generative and self-supervised models struggle with friction estimation. Performance again improves with domain adaptation. (iii) **MLLMs better on real test split than synthetic**. MLLMs continue to perform better on the real test split than synthetic test splits, benefiting from their familiarity with real-world visual semantics. (iv) **greater gap from oracle**. The performance gap between video foundation models and the oracle is more pronounced here than in the relative setting, indicating that accurate physical value regression remains a significant challenge for current video models.

Table 1: **Results for relative value comparison and absolute value prediction.** Left: ROC AUC scores for relative comparisons (range $[0, 1]$). Right: Pearson correlation coefficients for absolute predictions (range $[-1, 1]$). * indicates results after domain adaptation using a disjoint real training set. `test-1` is the synthetic in-distribution test split; `test-2` is the synthetic out-of-distribution test split; `test-3` is the real-world test split.

| Property | Method | Relative – ROC AUC | | | Absolute – Pearson Corr. | | |
|---|---|---|---|---|---|---|---|
| | | Test-1 | Test-2 | Test-3 | Test-1 | Test-2 | Test-3 |
| Elasticity | **Oracle** | 1.00 | 1.00 | 1.00 | 0.99 | 0.98 | 0.87 |
| | Video Generative Model | 1.00 | 0.98 | 0.84 | 0.92 | 0.82 | 0.07 |
| | Video Self-Supervised Model | 0.89 | 0.96 | 0.77 | 0.96 | 0.93 | 0.47 |
| | Qwen2.5VL-max | 0.59 | 0.50 | 0.54 | -0.05 | 0.11 | 0.16 |
| | GPT-4o | 0.51 | 0.66 | 0.62 | 0.19 | 0.11 | 0.30 |
| | Gemini-2.5-pro | 0.64 | 0.80 | 0.47 | 0.04 | 0.15 | 0.24 |
| Viscosity | **Oracle** | 0.99 | 1.00 | 1.00 | 0.99 | 0.98 | 0.80 |
| | Video Generative Model | 1.00 | 1.00 | 1.00 | 0.99 | 0.95 | 0.76 |
| | Video Self-Supervised Model | 1.00 | 1.00 | 0.99 | 1.00 | 0.97 | 0.79 |
| | Qwen2.5VL-max | 0.64 | 0.61 | 0.86 | 0.16 | 0.06 | 0.02 |
| | GPT-4o | 0.63 | 0.59 | 0.99 | 0.18 | 0.08 | 0.55 |
| | Gemini-2.5-pro | 0.48 | 0.69 | 0.95 | -0.06 | -0.05 | 0.60 |
| Friction | **Oracle** | 1.00 | 1.00 | 0.87 | 0.99 | 1.00 | 0.83 |
| | Video Generative Model | 0.98 | 0.89 | 0.47 | 0.95 | 0.78 | 0.21 |
| | + Domain Adaptation | – | – | 0.74* | – | – | 0.82* |
| | Video Self-Supervised Model | 1.00 | 0.97 | 0.58 | 0.71 | 0.58 | 0.28 |
| | + Domain Adaptation | – | – | 0.63* | – | – | 0.71* |
| | Qwen2.5VL-max | 0.50 | 0.62 | 0.80 | 0.03 | 0.14 | 0.06 |
| | GPT-4o | 0.34 | 0.42 | 0.67 | -0.10 | 0.03 | 0.38 |
| | Gemini-2.5-pro | 0.54 | 0.59 | 0.97 | -0.03 | -0.05 | 0.12 |

## 5.3 QUALITATIVE RESULTS

Figure 4 (left) shows qualitative examples comparing model predictions across different tasks. In the **first row**, we illustrate an example from the elasticity absolute value prediction task. The video generative model, self-supervised model, and MLLMs predict values of 0.83, 0.96, and 1.0, respectively—all reasonably close to the ground-truth value of 0.89. In the **second row**, we present a friction relative value comparison task. The input consists of two videos, where the first exhibits a higher dynamic friction coefficient than the second. Both the video generative model and the MLLM correctly assign high likelihoods to this relationship (0.82 and 1.0, respectively), aligning with the ground truth. In contrast, the self-supervised model incorrectly predicts the reverse and does so with high confidence. Figure 4 (right) shows examples of the scatter plots for the absolute value prediction. More specifically, we show the scatter plots of video self-supervised model on the three test splits. It can be observed that the performance degrades from `test-1` to `test-3`, as `test-1` is of the same distribution as the synthetic training split, while `test-2` is out-of-distribution synthetic test and `test-3` is for real evaluation. We provide more scatter plots in Supplementary Section F.

## 6 CONCLUSION

We investigate the task of inferring dynamic physical properties—elasticity, viscosity, and friction—from videos. To support this, we introduce a benchmark dataset with ground-truth annotations and evaluate a range of video foundation models under both absolute prediction and relative comparison settings. We adopt a simple architecture to extract physical cues from off-the-shelf generative and self-supervised video models, and explore prompting strategies to elicit predictions from MLLMs. Experiments show that generative and self-supervised models have similar performance. MLLMs perform worse overall but improve with more informative prompting, especially on real-world data. However, all models fall short of the oracle, particularly in absolute value prediction. These results highlight the need to enhance physical reasoning in video models—a key direction for future research.

**Reproducibility Statement.** To ensure the reproducibility of this work: (i) We have included the details of the dataset in Section 3.1 of the main paper and Section B of the supplementary. (ii) We have included the details of the method in Sections 4 and 5 of the main paper, and Sections A, C, D and E of the supplementary. (iii) We have included the code for our datasets and models in the supplementary.

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

# Supplementary Material

Besides this PDF file, the other files included in the supplementary are in the .zip folder:

- Folder 'sample_videos' - video samples of our *PhysVid* dataset
- Folder 'models' - code for Video Generative Model, Video Self-Supervised Model and MLLM
- Folder 'data' - code for generating synthetic datasets via Genesis

# A   ADDITIONAL IMPLEMENTATION DETAILS

As mentioned in Section 5 of the main paper, in this section we provide more implementation details. Please refer to our code in the .zip file for more details.

## A.1   COMPUTING INFRASTRUCTURE

The video generative and video self-supervised models are trained on a single H100/A6000/A40 GPU. The models are trained much faster on H100, and much slower on A6000 and A40, but all of them can be used to train and inference our models. The GPU memory of H100 GPU is 96GB, and 48GB for A6000/A40 GPU. The CPU memory associated with the GPU is 120GB for the H100 GPU, and 90GB for A6000/A40 GPU. The experiments are conducted on a Linux cluster. We provide the names and versions of the relevant software libraries and frameworks in the code.

## A.2   DIMENSION OF VIDEO FEATURE EXTRACTION VECTORS AND DETAILS OF MLPS

We set the dimension of the learnable query vector $q$ to be 2560 for DynamiCrafter or V-JEPA-2. For DynamiCrafter, $q$ is mapped to the dimensions of different layers for cross-attention with different layers of pre-trained feature tokens, by a set of MLPs $f_1^G : \mathbb{R}^{2560} \rightarrow \mathbb{R}^{(320 \times 4, 640 \times 3, 1280 \times 12, 640 \times 3, 320 \times 3)}$; For V-JEPA-2, $q$ is mapped to the dimensions of different layers for cross-attention with different layers of pre-trained feature tokens, by a set of MLPs $f_1^S : \mathbb{R}^{2560} \rightarrow \mathbb{R}^{(1024 \times 16)}$. The resulting vectors $p$ therefore can be represented as $\mathbb{R}^{(320 \times 4, 640 \times 3, 1280 \times 12, 640 \times 3, 320 \times 3)}$ for DynamiCrafter and $\mathbb{R}^{(1024 \times 16)}$ for V-JEPA-2. The $p$ vectors are then mapped by another set of MLPs $f_2^G : \mathbb{R}^{(320 \times 4, 640 \times 3, 1280 \times 12, 640 \times 3, 320 \times 3)} \rightarrow \mathbb{R}^{(2560 \times 25)}$ for DynamiCrafter, and $f_2^S : \mathbb{R}^{(1024 \times 16)} \rightarrow \mathbb{R}^{(2560 \times 16)}$ for V-JEPA-2 and then average pooled to get $P \in \mathbb{R}^{2560}$.

## A.3   DETAILS OF VIDEO INPUT

We uniformly sample 16 frames per video as input to all the models for fair comparison. The 16 frames are uniformly sampled so that the physics process we want to study is properly reflected with the sampled 16 frames, *e.g.,* the dropping and bouncing of the ball, the expansion of the liquid, and the slowing-down sliding process of the object. For construction of relative pairs of videos, we randomly get a list of different viewpoints first, and then for each viewpoint, we randomly generate $m$ videos and then randomly sample pairs from $m \times (m-1)$ possible video pairs. The binary ground truth for the pair is obtained via comparing the property values of the two videos.

## A.4   MOTIVATION AND DETAILS OF EVALUATION METRICS

We use the ROC AUC score for the relative formulation, as it is a binary classification problem, and AUC is a good evaluation metric to reflect the model's performance over different decision thresholds. ROC AUC is computed as the area under the receiver-operating-characteristic curve, *i.e.*, the integral of the true-positive rate versus the false-positive rate over all possible classification thresholds. We use the Pearson Correlation Coefficient for the absolute formulation, as it can reflect how the model's predicted values correlate to the ground truth values – and our goal here is the *value ordering*, rather than their absolute prediction, as accurately determining the ground truth values is difficult, particularly for viscosity. Also, the correlation is more forgiving than absolute error for out of distribution prediction (between the train and test sets).

## A.5   SEGMENTATION MASKS AND CORNER DETECTIONS FOR ORACLE ESTIMATION

In the oracle estimation, we need to segment the target object or liquid. For the synthetic videos, the segmentation masks can be directly obtained from the simulator. For the real videos, we obtain the segmentations by using the pre-trained Grounded SAM 2 (Ravi et al., 2025; Liu et al., 2024a; Ren et al., 2024a;b; Kirillov et al., 2023), with text prompts such as 'falling ball' (for elasticity), and

'sliding' + the names of different sliding objects (for friction). For viscosity, we only need the mask of the liquid on the plate surface, but the presence of the liquid column interferes – and directly using the liquid's name as a prompt for Grounded SAM 2 makes segmentation difficult. Therefore, we first segment the plate, and then apply morphological processing (taking the enclosed central region of the plate and using a closing operation to remove the liquid columns) to obtain the mask of the liquid on the plate surface. After getting the automatically predicted masks, we manually filter and adjust them to make sure they are of good quality. Apart from segmentation masks, for the friction oracle estimation, we also need to detect the corner points of the sliding cube. For the synthetic videos, we annotate the corners with a different color so we can easily detect them. For the real videos, we annotated the corner positions manually as they are difficult to obtain automatically.

## B   DATASET DETAILS

As mentioned in Section 3 of the main paper, in this section we provide further details regarding the collection of synthetic and real datasets.

### B.1   SYNTHETIC DATASETS

As described in Section 3 of the main paper, the simulator uses two distinct domains of nuisance parameters: $\mathcal{A}_1$ and $\mathcal{A}_2$. The `train` and `test-1` splits are generated by sampling from $\mathcal{A}_1$, while `test-2` is generated from $\mathcal{A}_2$. Below, we detail the differences between these domains for each dynamic physical property.

The Genesis simulator operates in a world coordinate system where gravity points in the $-z$ direction, and physical processes are centered around the origin $(0, 0, 0)$. The camera position is defined by three parameters: height $h$ (controls the $z$ coordinate), radius $R$ (distance from the $(0, 0)$ point in the $xy$-plane), and angle $\alpha$ (deviation from the $+x$ direction). Camera orientation is further specified by the 3D point $(x_l, y_l, z_l)$ that the camera looks at. Object and liquid colors are defined by RGB values $(r, g, b)$. Lighting remains fixed from the $+x$ direction, meaning changes in camera viewpoint also affect lighting conditions on the object.

Table 2, Table 3 and Table 4 detail the parameter settings for each physical property. All parameter ranges are chosen to ensure the visibility of the studied phenomena—e.g., the drop-and-bounce motion of a ball—in the synthetic videos.

Table 2: **Parameter Ranges for Elasticity**. Values are randomly sampled per domain if it is a range. Top: Nuisance parameters; Bottom: The target dynamic physical property we study.

| Parameter | $\mathcal{A}_1$ | $\mathcal{A}_2$ |
|---|---|---|
| $R$ | 1.5 | 1.5 |
| $h$ | $(0.5, 1.5)$ | $(0.25, 0.5)$ |
| $\alpha$ | $(0, \frac{1}{2}\pi)$ | $(\frac{1}{2}\pi, 2\pi)$ |
| $x_l$ | $(-0.1, 0.1)$ | $(0.1, 0.2)$ |
| $y_l$ | $(-0.1, 0.1)$ | $(-0.2, -0.1)$ |
| $z_l$ | $(0.05, 0.27)$ | $(-0.05, 0.05)$ |
| $r$ | $(0, 1)$ | $0$ |
| $g$ | $(0, 1)$ | $0$ |
| $b$ | $0$ | $(0, 1)$ |
| Drop height | $(0.25, 0.4)$ | $(0.4, 0.5)$ |
| Ball radius | 0.1 | |
| Elasticity | $(0, 1)$ | |

### B.2   REAL DATASETS DETAILS

**Elasticity Dataset.** This dataset contains video clips sourced from the Internet, capturing a variety of ball types being dropped—*e.g.*, basketball, tennis ball, soccer ball, rubber ball, balloon (air-filled), exercise ball, medicine ball, marble, and tomato. Ground truth elasticity values range from 0.44 (tomato) to 0.98 (tennis ball).

**Viscosity Dataset.** We include 12 different liquids: coffee, vinegar, cola, wine, cooking wine, whole milk, hot chocolate, dark soy sauce, smoothie, sesame oil, cream, and maple syrup. Ground truth viscosity values are obtained from online sources. For cases where the value is reported as a range, we use the midpoint as the ground truth. Values range from 1.2 (coffee) to 225 (maple syrup).

**Friction Dataset.** This dataset contains 5 objects—plastic disk, plastic LEGO brick, paper box, metal pencil box, and wooden box—and 6 surfaces: gray towel, kitchen paper, tablecloth, red towel, wooden table, and cardboard.

The training split includes:

Table 3: **Parameter Ranges for Viscosity**. Values are randomly sampled per domain if it is a range. Top: Nuisance parameters; Bottom: The target dynamic physical property we study.

| Parameter | $\mathcal{A}_1$ | $\mathcal{A}_2$ |
|---|---|---|
| $R$ | 1.5 | 1.5 |
| $h$ | $(0.5, 1.5)$ | $(0.25, 0.5)$ |
| $\alpha$ | $(0, \frac{1}{2}\pi)$ | $(\frac{1}{2}\pi, 2\pi)$ |
| $x_l$ | $(-0.1, 0.1)$ | $(0.1, 0.2)$ |
| $y_l$ | $(-0.1, 0.1)$ | $(-0.2, -0.1)$ |
| $z_l$ | $(0.05, 0.27)$ | $(-0.05, 0.05)$ |
| $r$ | $(0, 1)$ | 0 |
| $g$ | $(0, 1)$ | 0 |
| $b$ | 0 | $(0, 1)$ |
| Drop height | 0.056 | |
| Liquid column height | 0.1 | |
| Liquid column radius | 0.05 | |
| Viscosity | $(5e{-}5, 1e{-}2)$ | |

Table 4: **Parameter Ranges for Friction**. Values are randomly sampled per domain if it is a range. Top: Nuisance parameters; Bottom: The target dynamic physical property we study. $(x_0, y_0)$: initial position of the sliding cube; $(v_0^x, v_0^y)$: initial velocity of the sliding cube.

| Parameter | $\mathcal{A}_1$ | $\mathcal{A}_2$ |
|---|---|---|
| $R$ | 1.5 | 1.5 |
| $h$ | $(0.5, 1.5)$ | $(0.25, 0.5)$ |
| $\alpha$ | $(0, \frac{1}{2}\pi)$ | $(\frac{1}{2}\pi, 2\pi)$ |
| $x_l$ | $(-0.1, 0.1)$ | $(0.1, 0.2)$ |
| $y_l$ | $(-0.1, 0.1)$ | $(-0.2, -0.1)$ |
| $z_l$ | $(-0.1, 0.12)$ | $(-0.14, -0.1)$ |
| $r$ | $(0, 1)$ | $(0, 1)$ |
| $g$ | $(0, 1)$ | $(0, 1)$ |
| $b$ | 0 | $(0, 1)$ |
| $x_0$ | $(-0.1, 0.1)$ | $(-0.15, -0.1)$ |
| $y_0$ | $(-0.1, 0.1)$ | $(0.1, 0.15)$ |
| $v_0^x$ | $(0.6, 1.0)$ | $(1.0, 1.2)$ |
| $v_0^y$ | $(0.6, 1.0)$ | $(1.0, 1.2)$ |
| Motion direction | $v_0^x$ or $v_0^y$ with prob. 0.5 | |
| Cube size | 0.1 | |
| Friction coeff. | $(0, 0.2)$ | |

- **Objects:** paper box, metal pencil box, wooden box

- **Surfaces:** red towel, wooden table, cardboard

The testing split includes:

- **Objects:** plastic disk, LEGO brick

- **Surfaces:** gray towel, kitchen paper, tablecloth

Figure 5 shows close-up images of all objects and surfaces. Ground truth friction values range from 0.105 (pencil box on plastic paperboard) to 0.544 (LEGO on gray towel). The ground truth dynamic friction coefficient values are measured using a spring dynamometer by dragging the object at constant speed, as mentioned in the main paper. All datasets will be made publicly available upon paper acceptance under the CC-BY-4.0 license.

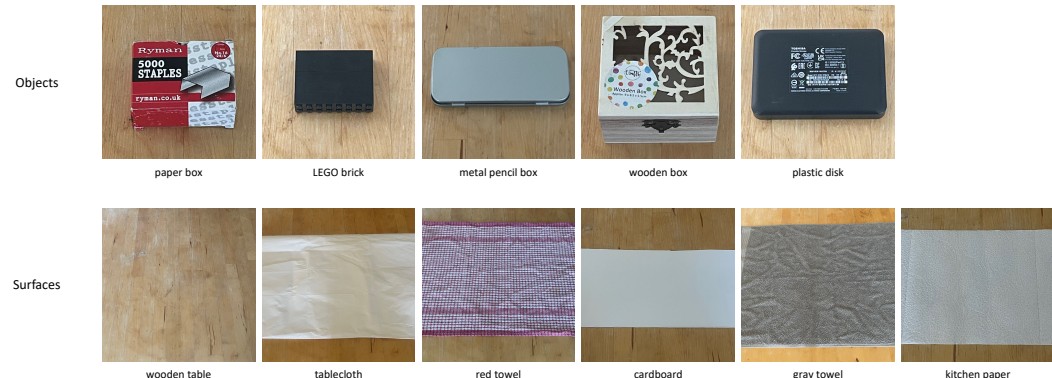

Figure 5: **Objects and surfaces in the *friction* real dataset.** Top: Objects used for friction real dataset collection; Bottom: Surfaces used for friction real dataset collection.

## B.3 DEVICES FOR REAL DATASET COLLECTION

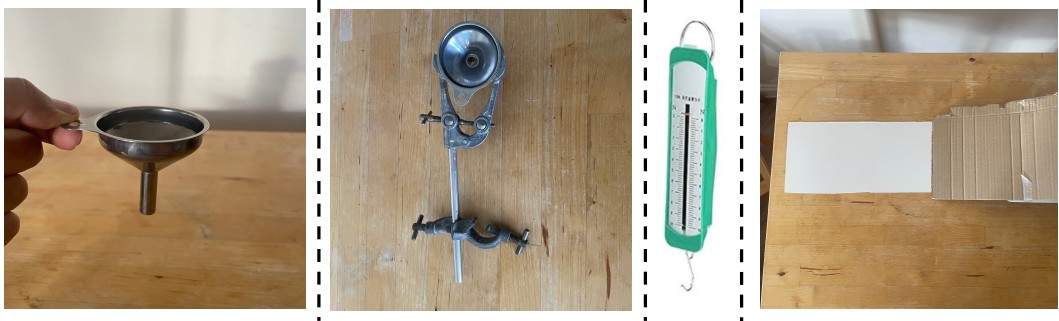

Figure 6: **Devices used to collect real datasets. Left**: The funnel used in the collection of the viscosity real dataset; **Middle-left**: The funnel holder used in the collection of the viscosity real dataset; **Middle-right**: The spring dynamometer used to measure the ground truth dynamic friction coefficient in the collection of the friction real dataset; **Right**: The slope used to give the objects an initial velocity on the horizontal surface in the collection of the friction real dataset.

As mentioned in Section 3.1 of the main paper, we show the devices that we used to collect our real datasets in Figure 6. The funnel (left) and the funnel holder (middle-left) are used to collect the viscosity real dataset. The spring dynamometer (middle-right) is used to measure the ground truth dynamic friction coefficient in the collection of the friction real dataset. The slope (right) is used in the friction experiment, where we slides the object down the slope to give it an initial velocity on the horizontal surface.

## C  DERIVATION OF ORACLE ESTIMATIONS EQUATIONS

As mentioned in Section 3.1 of the main paper, in this section, we give the derivation for the oracle estimations in the main paper, based on standard definitions for the properties and the laws of physics. Note, we only require the oracle values for each property to be determined up to a common scale, since the correlation used to evaluate performance is unaffected by the overall scale.

**Elasticity.** The *coefficient of elasiticity* or *coefficient of restitution* $e$ is defined as

$$e = \frac{v_{\text{before impact}}}{v_{\text{after impact}}}$$

where $v$ is the magnitude of the velocity. In our case the ball is dropped from rest at a height $h_{\text{drop}}$ onto a horizontal surface, and bounces in the vertical direction to the height $h_{\text{bounce}}$. It can be shown (see Wikipedia contributors (2025a)) that in this case:

$$e = \sqrt{\frac{h_{\text{bounce}}}{h_{\text{drop}}}} \tag{2}$$

**Viscosity.**  *Viscosity* is a physical property that characterizes a fluid's internal resistance to flow (Wikipedia contributors, 2025b). In our case, we study the case that the liquid is expanding on the ground plane. According to the spreading dynamics (Wikipedia contributors, 2025c) of liquid, the radius (thus the area) of the liquid is an inverse function of the viscosity, given other parameters controlled, such as the density of the liquid $\rho$, the diameter (thus the volume) of the liquid column $D$, and the dropping velocity $v$ of the liquid column when it reaches the ground. In our case, we control $D$ as we use a funnel with a fixed nozzle diameter to produce a consistent liquid column and we always pour the same volume of liquid into the funnel; we control $v$ as we use a funnel holder that allows us to fix the height from which the liquid is poured; $\rho$ is roughly controlled as the liquids are of similar density, ranging from $0.92$ (*e.g.,* sesame oil ) to $1.05$ (*e.g.,* smoothie), except for maple syrup which is $1.32$, but as the ground truth viscosity of maple syrup is much higher than other liquids, this variation will not influence much when we calculate the Pearson Correlation Coefficient between predictions and ground truth values. Therefore, we assume

$$\mu \propto \frac{1}{(d(A(t))/dt)^\alpha} \tag{3}$$

where $\mu$ is the viscosity of the liquid, and $A(t)$ is the liquid area size as a function of time $t$. In practice, we try with $\alpha = 1$, *i.e.,* $\mu \propto \frac{1}{d(A(t))/dt}$ and gets reasonable oracle test results, so we set $\alpha = 1$ for our oracle estimations.

**Friction.** If $F$ is the dynamic friction force acting on the object, then the dynamic friction coefficient $\mu_k$ is defined by the equation $F = \mu_k \times$ normal force on the object. In our case, the object moves on a horizontal surface, and the normal force is the weight of the object, so $F = \mu_k mg$, where $m$ is the mass of the object, and $g$ is the gravity acceleration. From Newton's Second Law $F = ma$, we therefore have $a = \mu_k g$, *i.e.,*

$$\mu_k = \frac{a}{g} \tag{4}$$

where $a$ is the acceleration of the object.

# D  ABLATION FOR DIFFERENT STRATEGIES OF MLLM PROMPTING

As mentioned in Section 5 of the main paper, we conduct an ablation study on the elasticity task to identify the most effective prompting strategy for MLLMs, using **Gemini 2.5 Pro** due to its strong performance in video understanding and visual reasoning. Results for the absolute and relative formulations are shown in Table 5 and Table 6, respectively. Due to the high computational cost of MLLM inference, we perform the ablation on a randomly selected subset of 20 samples per test split. The results show that the **Few-Shot Examples** strategy performs best for the absolute formulation, while **Oracle Estimation Teaching** is most effective for the relative formulation.

In Section E, we provide examples for each of the prompting strategies and detailed analysis regarding the influence of each strategy to the final performance.

Table 5: **Absolute prediction results for different MLLM prompting strategies**. We conduct the study on Gemini 2.5 Pro for the elasticity task.

| Strategy | Test-1 | Test-2 | Test-3 | Avg |
|---|---|---|---|---|
| Baseline | -0.03 | 0.26 | 0.06 | 0.10 |
| + Frame Index Provided | 0.06 | 0.35 | 0.55 | 0.32 |
| + Few-Shot Examples | 0.39 | 0.34 | 0.24 | **0.33** |
| + Oracle Estimation Teaching | 0.19 | 0.14 | 0.24 | 0.19 |

Table 6: **Relative comparison results for different MLLM prompting strategies**. We conduct the study on Gemini 2.5 Pro for the elasticity task.

| Strategy | Test-1 | Test-2 | Test-3 | Avg |
|---|---|---|---|---|
| Baseline | 0.51 | 0.74 | 0.55 | 0.60 |
| + Black Frames in Between | 0.56 | 0.72 | 0.63 | 0.64 |
| + Frame Index Provided | 0.54 | 0.80 | 0.52 | 0.62 |
| + Few-Shot Examples | 0.43 | 0.65 | 0.52 | 0.54 |
| + Oracle Estimation Teaching | 0.63 | 0.79 | 0.54 | **0.65** |

# E    EXAMPLES OF DIFFERENT PROMPTING STRATEGIES

As mentioned in Section 4.3 of the main paper, in this section we provide examples of different prompting strategies for both the **absolute formulation** and the **relative formulation**.

The examples of **absolute formulation** are provided in Figure 7 to Figure 11. More specifically, Figure 7 shows the visual input to the MLLM; Figure 8 shows the prompt and model output of *baseline prompt*; Figure 9 shows the prompt and model output of *oracle estimation teaching*; Figure 10 shows the prompt and model output of *few-shot examples*; Figure 11 shows the prompt and model output of *frame index provided*.

It can be observed that:

- **Baseline Prompt.** The initial state of object motion is incorrectly recognized from the beginning.
- **Oracle Estimation Teaching.** Although the model strictly follows the oracle's step-by-step guidance, an incorrect identification of the peak in the third step leads to a significantly inaccurate final prediction.
- **Few-Shot Examples.** The ground-truth examples provided in the few-shot setting serve as effective calibration signals, leading to notably improved performance.
- **Frame Index Provided.** Providing frame indices helps the model better interpret the motion process. However, estimating the final value based solely on this information remains challenging.

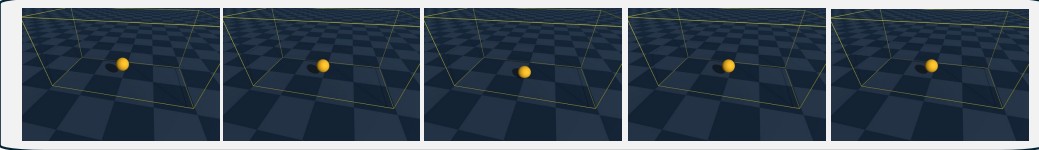

Figure 7: **Visual input for the *absolute formulation*.** The example is on Gemini for the elasticity property. Five representative frames are shown for the input video.

```
system_prompt = {

  'role': 'system', 'content': 'You are an expert in elasticity analysis with the ability to analyze
  video frames and determine elasticity values. You MUST provide a specific elasticity estimate.
  Never say you cannot analyze the images.'

  'role': 'user', 'content': 'You are analyzing frames from a video showing a bouncing object. Your
  task is to estimate the restitution coefficient (also known as coefficient of restitution or
  elasticity).

  The restitution coefficient is the ratio of separation speed to approach speed during collision:
  - Perfectly elastic collision: 1.0 (ball bounces back with same speed)
  - Perfectly inelastic collision: 0.0 (ball sticks to surface, no bounce)
  - Typical values: 0.1-0.9 for most materials

  If the object bounces back significantly → restitution coefficient closer to 1.0
  If the object barely bounces or sticks → restitution coefficient closer to 0.0

  YOUR RESPONSE MUST END WITH A SPECIFIC RESTITUTION COEFFICIENT VALUE IN THIS FORMAT:
  "Estimated restitution coefficient value: X "

  Where X is a number between 0 and 1 (e.g., 0.7, 0.3, 0.85).' }

visual_input = {
"type": "image_url", "image_url": {"url": f"data:image/jpeg;base64,{frame1}"}
"type": "image_url", "image_url": {"url": f"data:image/jpeg;base64,{frame2}"}
 ...
"type": "image_url", "image_url": {"url": f"data:image/jpeg;base64,{frame16}"}
}

[Output]

Based on my analysis of the provided frames, I will now estimate the restitution coefficient.

The frames show a yellow sphere suspended or resting within a defined space. There is no visible
downward motion or approach towards the surface in the sequence of images. The object remains
stationary. Consequently, there is no observable bounce or separation from the surface. The
object's vertical velocity after the point of potential impact is zero. Since the separation speed
is zero, this indicates a complete loss of kinetic energy upon impact, which is characteristic of a
perfectly inelastic collision. The object does not rebound at all.

Given that the object shows no bounce whatsoever, the separation velocity is zero.

Estimated restitution coefficient value: 0.0

[Ground-Truth]

GT value: 0.933
```

Figure 8: **Example of *baseline prompt* for the *absolute formulation*.** The example is on Gemini for the elasticity property. The initial state of object motion is incorrectly recognized from the beginning.

```
system_prompt = {

'role': 'system', 'content': 'You are an expert in elasticity analysis with the ability to analyze
video frames and determine elasticity values. You MUST provide a specific elasticity estimate.
Never say you cannot analyze the images.'

'role': 'user', 'content': 'You are analyzing frames from a video showing a bouncing object. Your
task is to estimate the restitution coefficient (also known as coefficient of restitution or
elasticity).

[Oracle Guidance]
ANALYSIS METHOD:
The intrinsic visual cue to reflect the elasticity property is the dropping height and bouncing
height of the ball. You only need to calculate the ratio of the height difference in the camera
coordinate. From this trajectory, you need to identify key points: the initial position, first
ground contact, and bounce peak.

Follow these steps:
1. Identify the ball's initial dropping position (highest point before impact)
2. Locate the first ground contact point (lowest point during impact)
3. Find the bounce peak (highest point after first bounce)
4. Calculate the height ratio: (bounce height - ground level) / (initial height - ground level)
5. This ratio represents the restitution coefficient (elasticity)

The restitution coefficient is the ratio of separation speed to approach speed during collision:
- Perfectly elastic collision: 1.0 (ball bounces back with same speed)
- Perfectly inelastic collision: 0.0 (ball sticks to surface, no bounce)
- Typical values: 0.1-0.9 for most materials

If the object bounces back significantly → restitution coefficient closer to 1.0
If the object barely bounces or sticks → restitution coefficient closer to 0.0

YOUR RESPONSE MUST END WITH A SPECIFIC RESTITUTION COEFFICIENT VALUE IN THIS FORMAT:
"Estimated restitution coefficient value: X "

Where X is a number between 0 and 1 (e.g., 0.7, 0.3, 0.85).' }

visual_input = {
"type": "image_url", "image_url": {"url": f"data:image/jpeg;base64,{frame1}"}
"type": "image_url", "image_url": {"url": f"data:image/jpeg;base64,{frame2}"}
...
"type": "image_url", "image_url": {"url": f"data:image/jpeg;base64,{frame16}"}
}

[Output]

Based on my analysis of the provided frames, I will now estimate the restitution coefficient.

1.  Initial Dropping Position: The initial dropping position is not explicitly shown, as the
ball is already on the ground in the first frame. I will assume it was dropped from a height
greater than its current position.

2.  Ground Contact Point: In all provided frames (`frame1` through `frame16`), the yellow sphere
is shown stationary and in contact with the ground surface. The lowest point of the trajectory
is the ground itself.

3.  Bounce Peak: The frames show no upward movement after the presumed impact. The ball remains
on the ground, indicating that the peak height of the bounce is zero.

4.  Height Ratio Calculation: Let the initial drop height be `H_initial`. Let the ground level
be `0`. The observed bounce height `H_bounce` is `0`. The height ratio is calculated as
`(H_bounce - ground level) / (H_initial - ground level)`. This gives a ratio of `0 / H_initial =
0`.

5.  Restitution Coefficient: A ratio of 0 indicates a perfectly inelastic collision. The object
impacted the surface and did not bounce back at all, losing all of its vertical kinetic energy.

Estimated restitution coefficient value: 0.0

[Ground-Truth]

GT value: 0.933
```

Figure 9: **Example of *oracle estimation teaching* for the *absolute formulation*.** The example is on Gemini for the elasticity property. Although the model strictly follows the oracle's step-by-step guidance, an incorrect identification of the peak in the third step leads to a significantly inaccurate final prediction.

```
system_prompt = {

'role': 'system', 'content': 'You are an expert in elasticity analysis with the ability to analyze
video frames and determine elasticity values. You MUST provide a specific elasticity estimate.
Never say you cannot analyze the images.'

'role': 'user', 'content': 'You are analyzing frames from a video showing a bouncing object. Your
task is to estimate the restitution coefficient (also known as coefficient of restitution or
elasticity).

The restitution coefficient is the ratio of separation speed to approach speed during collision:
- Perfectly elastic collision: 1.0 (ball bounces back with same speed)
- Perfectly inelastic collision: 0.0 (ball sticks to surface, no bounce)
- Typical values: 0.1-0.9 for most materials

If the object bounces back significantly → restitution coefficient closer to 1.0
If the object barely bounces or sticks → restitution coefficient closer to 0.0

YOUR RESPONSE MUST END WITH A SPECIFIC RESTITUTION COEFFICIENT VALUE IN THIS FORMAT:
"Estimated restitution coefficient value: X "

Where X is a number between 0 and 1 (e.g., 0.7, 0.3, 0.85).' }

[Few-shot Examples]

visual_input = {

"type": "image_url", "image_url": {"url": f"data:image/jpeg;base64,{example_frame1}"}
"type": "image_url", "image_url": {"url": f"data:image/jpeg;base64,{example_frame2}"}
...
"type": "image_url", "image_url": {"url": f"data:image/jpeg;base64,{example_frame16}"}
}

Output: Estimated restitution coefficient value: 0.750

Now analyze the following video:

visual_input = {
"type": "image_url", "image_url": {"url": f"data:image/jpeg;base64,{frame1}"}
"type": "image_url", "image_url": {"url": f"data:image/jpeg;base64,{frame2}"}
...
"type": "image_url", "image_url": {"url": f"data:image/jpeg;base64,{frame16}"}
}

[Output]

Based on my analysis of the provided video frames, I have determined the following:

1. Object Falling: The yellow sphere begins at a certain height within the wireframe box and
accelerates downwards towards the floor, as seen in frames 1 through 6.

2. Impact Moment: In frame 7, the sphere makes contact with the floor. A slight flattening at
the point of contact indicates the impact.

3. Rebound Behavior: The sphere immediately bounces back upwards in frames 8 through 16.

4. Speed Comparison: The sphere rebounds with significant energy. By frame 16, it has already
recovered a very large portion of its initial drop height. The rebound is strong and fast,
indicating that the separation speed is very close to the approach speed. This is a highly
elastic collision with minimal energy loss.

Estimated restitution coefficient value: 0.92

[Ground-Truth]

GT value: 0.933
```

Figure 10: **Example of *few-shot examples* for the *absolute formulation*.** The example is on Gemini for the elasticity property. The ground truth examples provided in the few-shot setting serve as effective calibration signals, leading to notably improved performance.

```
system_prompt = {

 'role': 'system', 'content': 'You are an expert in elasticity analysis with the ability to analyze
 video frames and determine elasticity values. You MUST provide a specific elasticity estimate.
 Never say you cannot analyze the images.'

 'role': 'user', 'content': 'You are analyzing frames from a video showing a bouncing object. Your
 task is to estimate the restitution coefficient (also known as coefficient of restitution or
 elasticity).

 The restitution coefficient is the ratio of separation speed to approach speed during collision:
 - Perfectly elastic collision: 1.0 (ball bounces back with same speed)
 - Perfectly inelastic collision: 0.0 (ball sticks to surface, no bounce)
 - Typical values: 0.1-0.9 for most materials

 If the object bounces back significantly → restitution coefficient closer to 1.0
 If the object barely bounces or sticks → restitution coefficient closer to 0.0

 YOUR RESPONSE MUST END WITH A SPECIFIC RESTITUTION COEFFICIENT VALUE IN THIS FORMAT:
 "Estimated restitution coefficient value: X "

 Where X is a number between 0 and 1 (e.g., 0.7, 0.3, 0.85).' }

[Frame Index Provided]

visual_input = {
{"type": "text", "text": "frame1:"}
"type": "image_url", "image_url": {"url": f"data:image/jpeg;base64,{frame1}"}
{"type": "text", "text": "frame2:"}
"type": "image_url", "image_url": {"url": f"data:image/jpeg;base64,{frame2}"}
...
{"type": "text", "text": "frame16:"}
"type": "image_url", "image_url": {"url": f"data:image/jpeg;base64,{frame16}"}
}

[Output]

Based on my analysis of the provided frames, I will now estimate the restitution coefficient.

The initial frames show the object already in contact with the ground surface. There is no visible
falling motion. The object is in contact with the surface from the very first frame. Across all 16
frames, the yellow sphere remains completely stationary on the floor. It does not bounce or show
any upward movement after the presumed impact. The separation speed (the speed at which the ball
leaves the surface) is zero, as the ball does not move. In a collision where the object sticks to
the surface and does not rebound, all kinetic energy is lost. This is characteristic of a perfectly
inelastic collision.

Estimated restitution coefficient value: 0.0

[Ground-Truth]

GT value: 0.933
```

Figure 11: **Example of *frame index provided* for the *absolute formulation*.** The example is on Gemini for the elasticity property. Providing frame indices helps the model better interpret the motion process. However, estimating the final value based solely on this information remains challenging.

The examples of **relative formulation** are provided in Figure 12 to Figure 17. More specifically, Figure 12 shows the visual input to the MLLM; Figure 13 shows the prompt and model output of *baseline prompt*; Figure 14 shows the prompt and model output of *oracle estimation teaching*; Figure 15 shows the prompt and model output of *few-shot examples*; Figure 16 shows the prompt and model output of *frame index provided*; Figure 17 shows the prompt and model output of *black frames in between*.

It can be observed that:

- **Baseline Prompt.** The baseline model exhibits reasonable performance.

- **Oracle Estimation Teaching.** The oracle strategy promotes qualitative analysis (*e.g.*, comparing motion or relative magnitudes) without forcing exact calculations. This flexible reasoning process leads to more reliable outputs.

- **Few-Shot Examples.** The relative task is simpler—determining which of two instances has a greater physical value—without requiring exact numerical estimates. Here, few-shot examples tend to degrade performance, often encouraging shortcut responses that reduce interpretability and stability.

- **Frame Index Provided.** Providing the frame indices enhances the model's understanding of temporal dynamics, thereby resulting in more effective comparative reasoning.

- **Black Frames in Between.** Concatenating both videos with black frames in between enables the model to better perform relative comparisons, likely by making inter-video relationships more explicit.

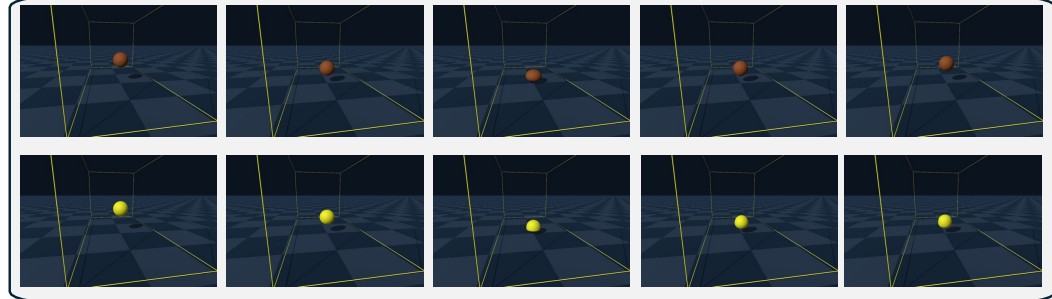

Figure 12: **Visual input for the *absolute formulation*.** The example is on Gemini for the elasticity property. Five representative frames are shown for the input videos. Top: the first video; Bottom: the second video. Ground truth should be 1 in this case as the first video shows a higher elasticity value than the second one.

```
system_prompt = {

    'role': 'system', 'content': 'You are an expert in elasticity analysis with the ability to compare
    video frames and determine which has higher elasticity values. You MUST provide a specific
    comparison result. Never say you cannot analyze the images.'

    'role': 'user', 'content': 'You are comparing two elasticity videos to determine which has a
    higher elasticity value.

    The first set of frames is labeled "VIDEO 1" and the second set is labeled "VIDEO 2". Compare the
    elasticity characteristics between the two videos: VIDEO 1 frames are shown first, VIDEO 2 frames
    are shown second.

    YOUR RESPONSE MUST END WITH A SPECIFIC CONFIDENCE SCORE IN THIS FORMAT: "Comparison result: X"

    Where X is a decimal number between 0.0 and 1.0:
    - X = 1.0: Very confident that VIDEO 1 has higher elasticity
    - X = 0.6: Somewhat confident that VIDEO 1 has higher elasticity
    - X = 0.5: Equal or uncertain (no clear difference)
    - X = 0.2: Quite confident that VIDEO 2 has higher elasticity
    - X = 0.0: Very confident that VIDEO 2 has higher elasticity

    The closer to 1.0, the more confident you are that VIDEO 1 has higher elasticity.
    The closer to 0.0, the more confident you are that VIDEO 2 has higher elasticity. ' }

    VIDEO 1= { "type": "image_url", "image_url": {"url": f"data:image/jpeg;base64,{frame1}"}
                ...
                "type": "image_url", "image_url": {"url": f"data:image/jpeg;base64,{frame16}"} }

    VIDEO 2= { "type": "image_url", "image_url": {"url": f"data:image/jpeg;base64,{frame1}"}
                ...
                "type": "image_url", "image_url": {"url": f"data:image/jpeg;base64,{frame16}"} }

    [Output]

    Based on my analysis of the provided video frames, I will compare the elasticity of the objects
    shown.

    In VIDEO 1, the brown ball is compressed from the top. It deforms by squashing downwards and then
    smoothly returns to its original shape once the compressive force is removed. This behavior is
    characteristic of a material with good elastic properties, as it fully recovers from the
    deformation.

    In VIDEO 2, the yellow ball is also compressed. However, during the compression, a visible fracture
    line appears on the side of the ball. Although the ball eventually returns to its original overall
    shape, the appearance of this fracture indicates that the material has undergone some form of
    failure or has exceeded its elastic limit. A purely elastic material would deform without
    fracturing. The presence of this crack, even if temporary, suggests that the material is less
    resilient and therefore less elastic than the one in the first video.

    Therefore, the object in VIDEO 1 demonstrates higher elasticity because it deforms and recovers
    without any signs of structural failure, whereas the object in VIDEO 2 shows signs of fracturing
    under stress.

    Comparison result: 0.8

    [Ground-Truth]

    GT value: 1
```

Figure 13: **Example of *baseline prompt* for the *relative formulation*.** The example is on Gemini for the elasticity property. The baseline model exhibits reasonable performance.

```
system_prompt = {

'role': 'system', 'content': 'You are an expert in elasticity analysis with the ability to compare
video frames and determine which has higher elasticity values. You MUST provide a specific
comparison result. Never say you cannot analyze the images.'

'role': 'user', 'content': 'You are comparing two elasticity videos to determine which has a
higher elasticity value.

[Oracle Guidance]
ANALYSIS METHOD:
The intrinsic visual cue to reflect the elasticity property is the dropping height and bouncing
height of the ball. You only need to calculate the ratio of the height difference in the camera
coordinate. From this trajectory, you need to identify key points: the initial position, first
ground contact, and bounce peak.

Follow these steps:
1. Identify the ball's initial dropping position (highest point before impact)
2. Locate the first ground contact point (lowest point during impact)
3. Find the bounce peak (highest point after first bounce)
4. Calculate the height ratio: (bounce height - ground level) / (initial height - ground level)
5. This ratio represents the restitution coefficient (elasticity)

The first set of frames is labeled "VIDEO 1" and the second set is labeled "VIDEO 2". Compare the
elasticity characteristics between the two videos: VIDEO 1 frames are shown first, VIDEO 2 frames
are shown second.

YOUR RESPONSE MUST END WITH A SPECIFIC CONFIDENCE SCORE IN THIS FORMAT: "Comparison result: X"

Where X is a decimal number between 0.0 and 1.0:
- X = 1.0: Very confident that VIDEO 1 has higher elasticity
- X = 0.6: Somewhat confident that VIDEO 1 has higher elasticity
- X = 0.5: Equal or uncertain (no clear difference)
- X = 0.2: Quite confident that VIDEO 2 has higher elasticity
- X = 0.0: Very confident that VIDEO 2 has higher elasticity

The closer to 1.0, the more confident you are that VIDEO 1 has higher elasticity.
The closer to 0.0, the more confident you are that VIDEO 2 has higher elasticity. ' }

VIDEO 1= { "type": "image_url", "image_url": {"url": f"data:image/jpeg;base64,{frame1}"}
           ...
           "type": "image_url", "image_url": {"url": f"data:image/jpeg;base64,{frame16}"} }

VIDEO 2= { "type": "image_url", "image_url": {"url": f"data:image/jpeg;base64,{frame1}"}
           ...
           "type": "image_url", "image_url": {"url": f"data:image/jpeg;base64,{frame16}"} }

[Output]

Based on the analysis of the provided video frames, I will determine the elasticity by calculating
the coefficient of restitution for each ball.

VIDEO 1 Analysis:

1. **Initial Dropping Position:** In the first frame, the ball is at its highest point before being
dropped. Let's denote its height as H_initial1.

2. **First Ground Contact:** In frame 6, the ball makes its first contact with the ground. We can set
this ground level as a height of 0.

3. **Bounce Peak:** After the impact, the ball rebounds. The peak of this first bounce appears to be
around frame 11. The height reached, H_bounce1, is visibly significant, approximately 40-50% of the
initial dropping height.

4. **Height Ratio (Elasticity):** The ratio (H_bounce1 / H_initial1) is roughly 0.4 to 0.5. The ball
retains a substantial amount of its energy, indicating a relatively high elasticity.

VIDEO 2 Analysis:

1. **Initial Dropping Position:** The ball starts from a similar height as in VIDEO 1. Let's call this
H_initial2.

2. **First Ground Contact:** The ball makes contact with the ground around frame 7. During contact
(frames 7-9), it deforms significantly, flattening out, which indicates a loss of energy through
deformation.

3. **Bounce Peak:** The ball pushes off the ground but barely gains any height. The peak of the bounce
(around frame 15) is very low, perhaps only 5-10% of the initial dropping height.

4. **Height Ratio (Elasticity):** The ratio (H_bounce2 / H_initial2) is very low, approximately 0.05
to 0.1. The significant deformation and minimal rebound show that most of the initial energy was
dissipated, indicating very low elasticity.

Comparison:

The ball in VIDEO 1 bounces back to a much greater height relative to its initial drop height
compared to the ball in VIDEO 2. The ball in VIDEO 2 loses almost all of its energy upon impact.
Therefore, the material in VIDEO 1 has a significantly higher elasticity.

Comparison result: 1.0

[Ground-Truth]

GT value: 1
```

Figure 14: **Example of *oracle estimation teaching* for the *relative formulation*.** The example is on Gemini for the elasticity property. The oracle strategy promotes qualitative analysis (*e.g.*, comparing motion or relative magnitudes) without forcing exact calculations. This flexible reasoning process leads to more reliable outputs.

```
system_prompt = {

'role': 'system', 'content': 'You are an expert in elasticity analysis with the ability to compare
video frames and determine which has higher elasticity values. You MUST provide a specific
comparison result. Never say you cannot analyze the images.'

'role': 'user', 'content': 'You are comparing two elasticity videos to determine which has a
higher elasticity value.

The first set of frames is labeled "VIDEO 1" and the second set is labeled "VIDEO 2". Compare the
elasticity characteristics between the two videos: VIDEO 1 frames are shown first, VIDEO 2 frames
are shown second.

YOUR RESPONSE MUST END WITH A SPECIFIC CONFIDENCE SCORE IN THIS FORMAT: "Comparison result: X"

Where X is a decimal number between 0.0 and 1.0:
- X = 1.0: Very confident that VIDEO 1 has higher elasticity
- X = 0.6: Somewhat confident that VIDEO 1 has higher elasticity
- X = 0.5: Equal or uncertain (no clear difference)
- X = 0.2: Quite confident that VIDEO 2 has higher elasticity
- X = 0.0: Very confident that VIDEO 2 has higher elasticity

The closer to 1.0, the more confident you are that VIDEO 1 has higher elasticity.
The closer to 0.0, the more confident you are that VIDEO 2 has higher elasticity. ' }

[Few-shot Examples]

VIDEO 1 = {"type": "image_url", "image_url": {"url": f"data:image/jpeg;base64,{example1_frame1}"}
            ...
            "type": "image_url", "image_url": {"url": f"data:image/jpeg;base64,{example1_frame16}"}}
VIDEO 2 = {"type": "image_url", "image_url": {"url": f"data:image/jpeg;base64,{example1_frame1}"}
            ...
            "type": "image_url", "image_url": {"url": f"data:image/jpeg;base64,{example1_frame16}"}}

Output: Comparison result: 0

Now analyze the following video pairs:

VIDEO 1= { "type": "image_url", "image_url": {"url": f"data:image/jpeg;base64,{frame1}"}
            ...
            "type": "image_url", "image_url": {"url": f"data:image/jpeg;base64,{frame16}"} }

VIDEO 2= { "type": "image_url", "image_url": {"url": f"data:image/jpeg;base64,{frame1}"}
            ...
            "type": "image_url", "image_url": {"url": f"data:image/jpeg;base64,{frame16}"} }

[Output]

Based on the frames provided:

- VIDEO 1 shows a object with consistent shape and size across frames, indicating limited
deformation.

- VIDEO 2 shows a similar object, but with more noticeable deformation and changes in shape.

The deformation in VIDEO 2 suggests higher elasticity compared to VIDEO 1.

Comparison result: 0.2

[Ground-Truth]

GT value: 1
```

Figure 15: **Example of *few-shot examples* for the *relative formulation*.** The example is on Gemini for the elasticity property. The relative task is simpler—determining which of two instances has a greater physical value—without requiring exact numerical estimates. Here, few-shot examples tend to degrade performance, often encouraging shortcut responses that reduce interpretability and stability.

```
system_prompt = {

    'role': 'system', 'content': 'You are an expert in elasticity analysis with the ability to compare
    video frames and determine which has higher elasticity values. You MUST provide a specific
    comparison result. Never say you cannot analyze the images.'

    'role': 'user', 'content': 'You are comparing two elasticity videos to determine which has a
    higher elasticity value.

    The first set of frames is labeled "VIDEO 1" and the second set is labeled "VIDEO 2". Compare the
    elasticity characteristics between the two videos: VIDEO 1 frames are shown first, VIDEO 2 frames
    are shown second.

    YOUR RESPONSE MUST END WITH A SPECIFIC CONFIDENCE SCORE IN THIS FORMAT: "Comparison result: X"

    Where X is a decimal number between 0.0 and 1.0:
    - X = 1.0: Very confident that VIDEO 1 has higher elasticity
    - X = 0.6: Somewhat confident that VIDEO 1 has higher elasticity
    - X = 0.5: Equal or uncertain (no clear difference)
    - X = 0.2: Quite confident that VIDEO 2 has higher elasticity
    - X = 0.0: Very confident that VIDEO 2 has higher elasticity

    The closer to 1.0, the more confident you are that VIDEO 1 has higher elasticity.
    The closer to 0.0, the more confident you are that VIDEO 2 has higher elasticity. ' }

VIDEO 1= { "type": "text", "text": "frame1:"
           "type": "image_url", "image_url": {"url": f"data:image/jpeg;base64,{frame1}"}
           ...
           "type": "text", "text": "frame16:"
           "type": "image_url", "image_url": {"url": f"data:image/jpeg;base64,{frame16}"} }

VIDEO 2= { "type": "text", "text": "frame1:"
           "type": "image_url", "image_url": {"url": f"data:image/jpeg;base64,{frame1}"}
           ...
           "type": "text", "text": "frame16:"
           "type": "image_url", "image_url": {"url": f"data:image/jpeg;base64,{frame16}"} }

[Output]

Based on my analysis of the provided video frames, I will now compare the elasticity
characteristics.

In VIDEO 1, the orange sphere impacts the ground and shows almost no visible deformation. It
maintains its spherical shape throughout the collision and rebounds to a height that is very close
to its initial drop height. This indicates a highly efficient transfer of energy and a very high
coefficient of restitution, which are key characteristics of high elasticity.

In VIDEO 2, the yellow sphere deforms significantly upon impact with the ground, flattening out
considerably before beginning to rebound. This large deformation suggests that a significant amount
of the kinetic energy is absorbed and dissipated as heat during the collision process. The
subsequent rebound appears less energetic than in VIDEO 1.

Therefore, the object in VIDEO 1 demonstrates a higher degree of elasticity, as it resists
deformation and conserves more kinetic energy through the collision.

Comparison result: 0.8

[Ground-Truth]

GT value: 1
```

Figure 16: **Example of *frame index provided* for the *relative formulation*.** The example is on Gemini for the elasticity property. Providing the frame indices enhances the model's understanding of temporal dynamics, thereby resulting in more effective comparative reasoning.

```
system_prompt = {

    'role': 'system', 'content': 'You are an expert in elasticity analysis with the ability to compare
    video frames and determine which has higher elasticity values. You MUST provide a specific
    comparison result. Never say you cannot analyze the images.'

    'role': 'user', 'content': 'You are comparing two elasticity videos to determine which has a
    higher elasticity value.

    The frames are arranged in sequence: VIDEO 1 frames first, followed by 3 BLACK SEPARATOR frames,
    then VIDEO 2 frames.

    YOUR RESPONSE MUST END WITH A SPECIFIC CONFIDENCE SCORE IN THIS FORMAT: "Comparison result: X"

    Where X is a decimal number between 0.0 and 1.0:
    - X = 1.0: Very confident that VIDEO 1 has higher elasticity
    - X = 0.6: Somewhat confident that VIDEO 1 has higher elasticity
    - X = 0.5: Equal or uncertain (no clear difference)
    - X = 0.2: Quite confident that VIDEO 2 has higher elasticity
    - X = 0.0: Very confident that VIDEO 2 has higher elasticity

    The closer to 1.0, the more confident you are that VIDEO 1 has higher elasticity.
    The closer to 0.0, the more confident you are that VIDEO 2 has higher elasticity. ' }

VIDEO = {  "type": "image_url", "image_url": {"url": f"data:image/jpeg;base64,{video1_frame1}"}
           ...
           "type": "image_url", "image_url": {"url": f"data:image/jpeg;base64,{video1_frame16}"}
           "type": "image_url", "image_url": {"url": f"data:image/jpeg;base64,{black_frame1}"}
           "type": "image_url", "image_url": {"url": f"data:image/jpeg;base64,{black_frame2}"}
           "type": "image_url", "image_url": {"url": f"data:image/jpeg;base64,{black_frame3}"}
           "type": "image_url", "image_url": {"url": f"data:image/jpeg;base64,{video2_frame1}"}
           ...
           "type": "image_url", "image_url": {"url": f"data:image/jpeg;base64,{video2_frame16}"} }

[Output]

Based on my analysis of the two videos, I will assess their elasticity.

The first video features a orange sphere. When dropped, it hits the ground and exhibits a
noticeable bounce. The sphere deforms slightly upon impact but quickly restores its original shape,
converting a significant portion of its potential energy into kinetic energy for the rebound. The
object continues to bounce multiple times, with each bounce decreasing in height, which is
characteristic of an elastic collision with some energy damping. The key takeaway is the
significant rebound and the tendency to maintain motion.

In VIDEO 2, the yellow sphere falls and hits the ground. It deforms dramatically, flattening out
almost completely against the surface. It then returns to its original spherical shape, but with a
low height, which results in a very low bounce. This significant deformation and subsequent
powerful rebound are clear indicators of low elasticity.

Comparing the two, the orange sphere in VIDEO 1 exhibits far greater elastic properties. It deforms
more and bounces back much more effectively, indicating a superior ability to store and release
energy during a collision.

Comparison result: 0.8

[Ground-Truth]

GT value: 1
```

Figure 17: **Example of *black frames in between* for the *relative formulation*.** The example is on Gemini for the elasticity property. Concatenating both videos with black frames in between enables the model to better perform relative comparisons, likely by making inter-video relationships more explicit.

## F  ADDITIONAL SCATTER PLOTS

As mentioned in Section 5.3 of the main paper, in this section we provide more scatter plots for different models on different test splits of the three dynamic physical properties.

Figure 18 shows the scatter plots of oracle estimation on different test splits of the three dynamic physical properties.

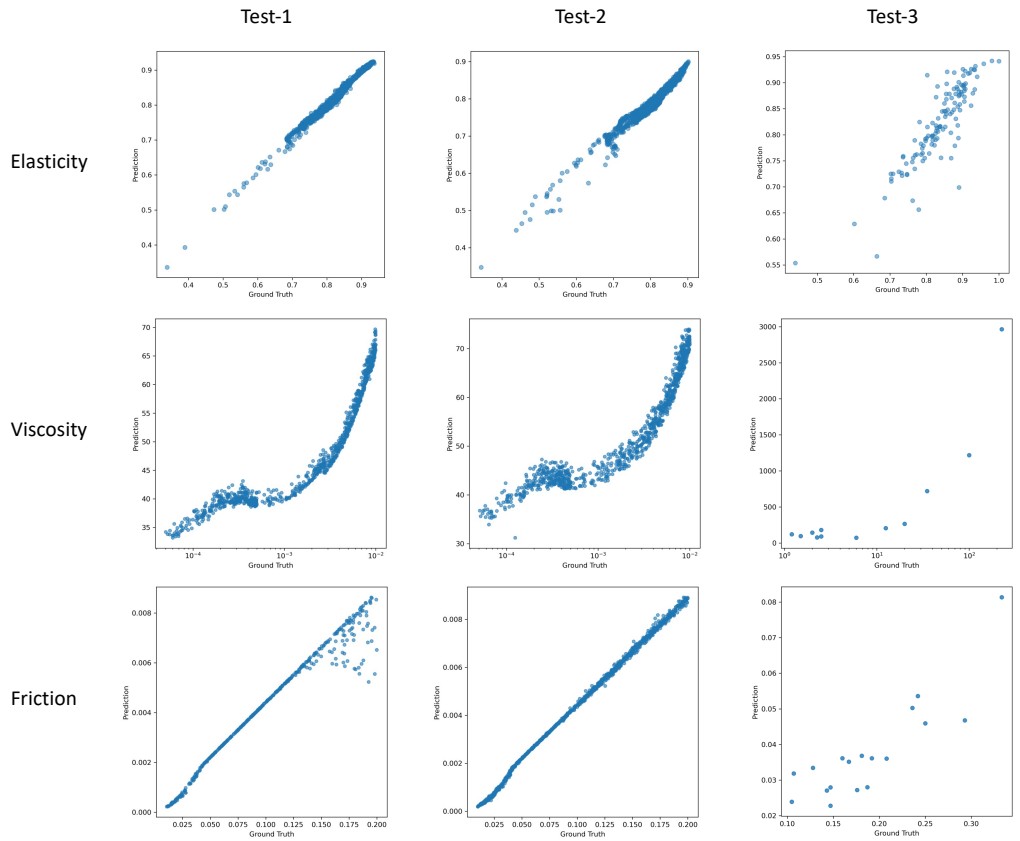

Figure 18: **Scatter plots for Oracle Estimation.** Top Row: Elasticity; Middle Row: Viscosity; Bottom Row: Friction. Left Column: Test-1; Middle Column: Test-2; Right Column: Test-3. For viscosity test-3, for each liquid, we take an average of the predictions for all samples to make the scatter plot, so that we can reduce the noise introduced by a single pouring liquid experiment; For friction test-3, for each combination of object and surface, we take an average of the predictions for all samples to make the scatter plot, so that we can reduce the noise introduced by a single sliding object experiment.

Figure 19 shows the scatter plots of DynamiCrafter on different test splits of the three dynamic physical properties.

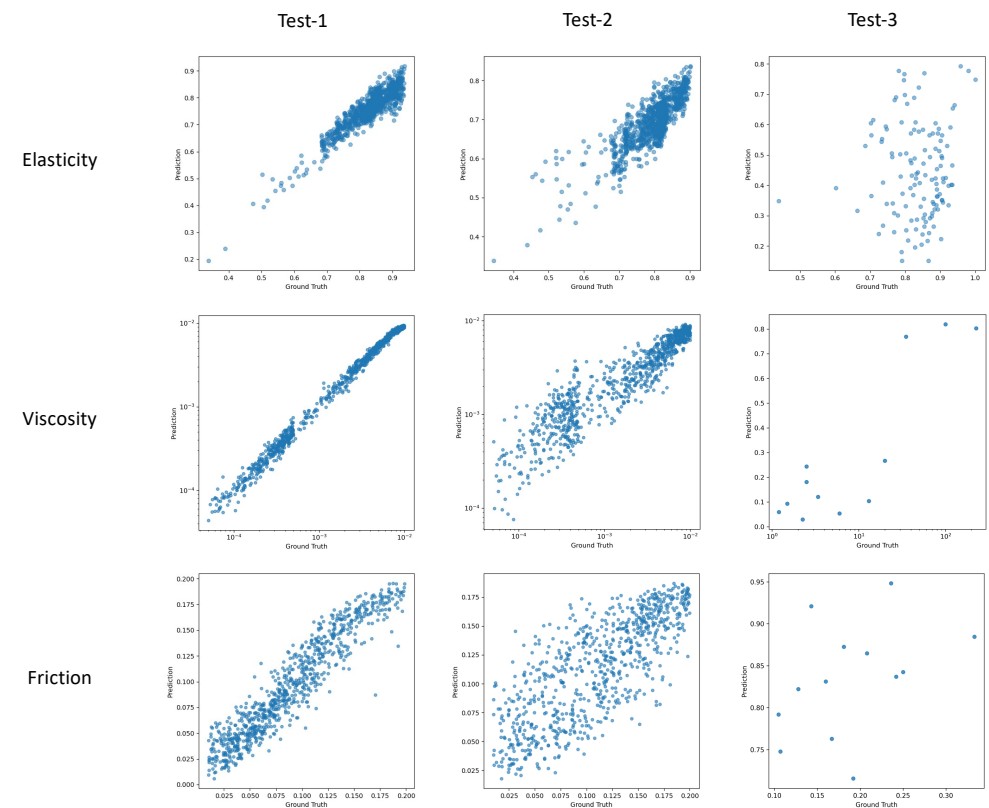

Figure 19: **Scatter plots for Video Generative Model.** Top Row: Elasticity; Middle Row: Viscosity; Bottom Row: Friction. Left Column: Test-1; Middle Column: Test-2; Right Column: Test-3. For viscosity `test-3`, for each liquid, we take an average of the predictions for all samples to make the scatter plot, so that we can reduce the noise introduced by a single pouring liquid experiment; For friction `test-3`, for each combination of object and surface, we take an average of the predictions for all samples to make the scatter plot, so that we can reduce the noise introduced by a single sliding object experiment.

Figure 20 shows the scatter plots of V-JEPA-2 on different test splits of the three dynamic physical properties.

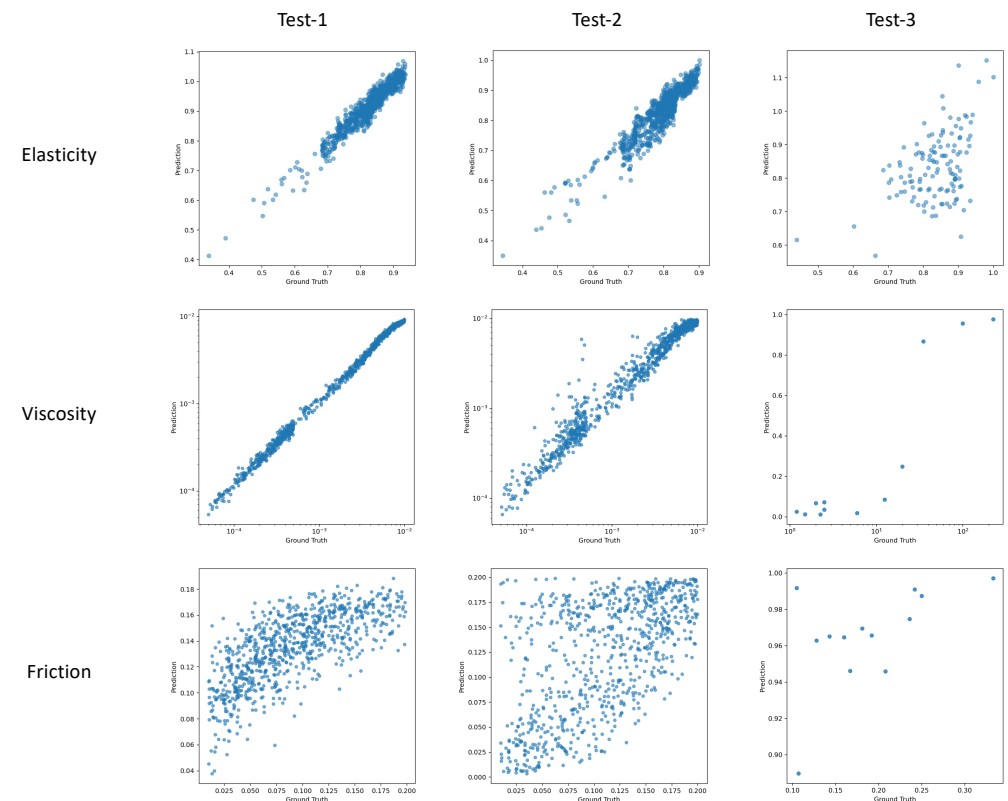

Figure 20: **Scatter plots for Video Self-Supervised Model.** Top Row: Elasticity; Middle Row: Viscosity; Bottom Row: Friction. Left Column: Test-1; Middle Column: Test-2; Right Column: Test-3. For viscosity `test-3`, for each liquid, we take an average of the predictions for all samples to make the scatter plot, so that we can reduce the noise introduced by a single pouring liquid experiment; For friction `test-3`, for each combination of object and surface, we take an average of the predictions for all samples to make the scatter plot, so that we can reduce the noise introduced by a single sliding object experiment.

Figure 21 shows the scatter plots of Qwen2.5VL-max on different test splits of the three dynamic physical properties. For `test-1` and `test-2`, due to the limitation of resources, a random subset of 100 samples are used.

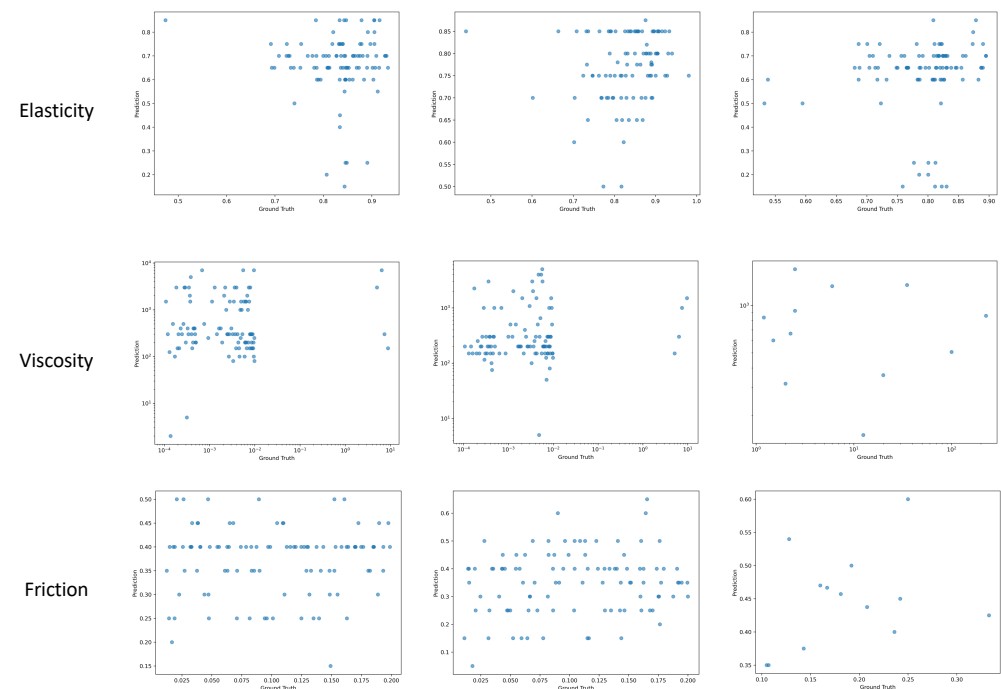

Figure 21: **Scatter plots for MLLMs (Qwen2.5VL-max).** Top Row: Elasticity; Middle Row: Viscosity; Bottom Row: Friction. Left Column: Test-1; Middle Column: Test-2; Right Column: Test-3. For viscosity `test-3`, for each liquid, we take an average of the predictions for all samples to make the scatter plot, so that we can reduce the noise introduced by a single pouring liquid experiment; For friction `test-3`, for each combination of object and surface, we take an average of the predictions for all samples to make the scatter plot, so that we can reduce the noise introduced by a single sliding object experiment.

Figure 22 shows the scatter plots of GPT-4o on different test splits of the three dynamic physical properties. For `test-1` and `test-2`, due to the limitation of resources, a random subset of 100 samples are used.

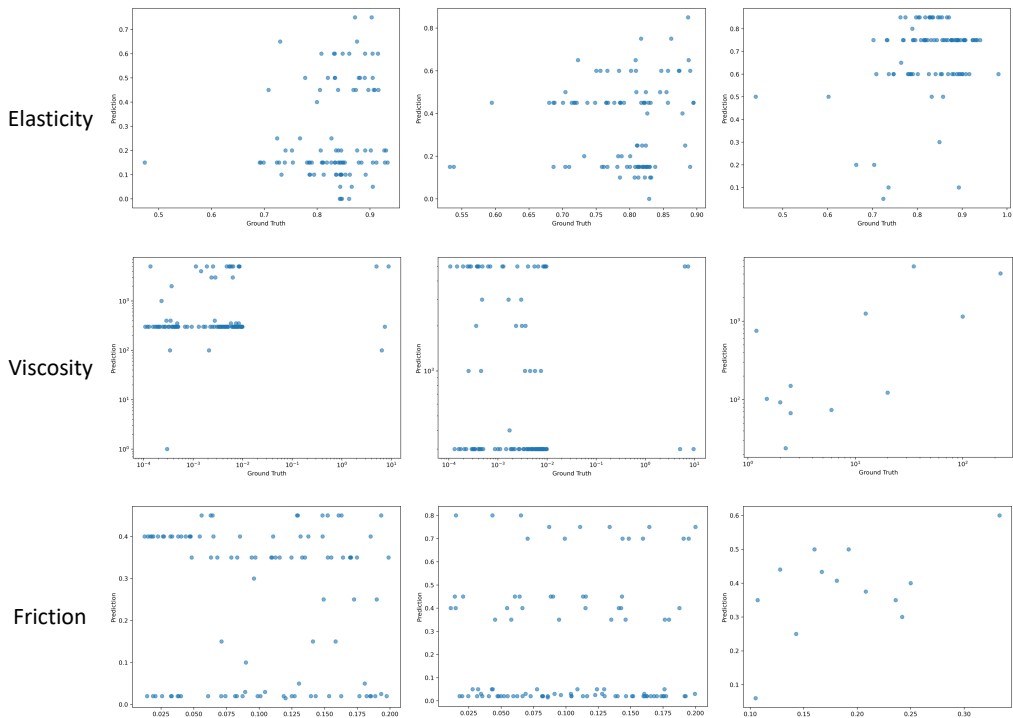

Figure 22: **Scatter plots for MLLMs (GPT-4o).** Top Row: Elasticity; Middle Row: Viscosity; Bottom Row: Friction. Left Column: Test-1; Middle Column: Test-2; Right Column: Test-3. For viscosity `test-3`, for each liquid, we take an average of the predictions for all samples to make the scatter plot, so that we can reduce the noise introduced by a single pouring liquid experiment; For friction `test-3`, for each combination of object and surface, we take an average of the predictions for all samples to make the scatter plot, so that we can reduce the noise introduced by a single sliding object experiment.

Figure 23 shows the scatter plots of Gemini-2.5-pro on different test splits of the three dynamic physical properties. For `test-1` and `test-2`, due to the limitation of resources, a random subset of 100 samples are used.

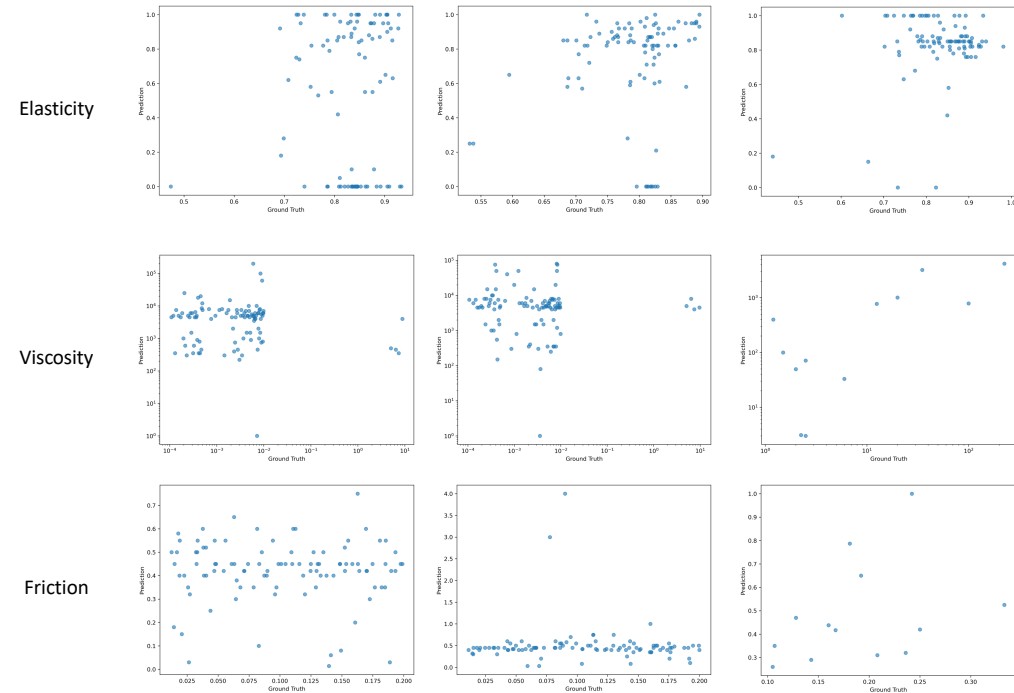

Figure 23: **Scatter plots for MLLMs (Gemini-2.5-pro).** Top Row: Elasticity; Middle Row: Viscosity; Bottom Row: Friction. Left Column: Test-1; Middle Column: Test-2; Right Column: Test-3. For viscosity `test-3`, for each liquid, we take an average of the predictions for all samples to make the scatter plot, so that we can reduce the noise introduced by a single pouring liquid experiment; For friction `test-3`, for each combination of object and surface, we take an average of the predictions for all samples to make the scatter plot, so that we can reduce the noise introduced by a single sliding object experiment.

## G    EFFECTIVENESS OF RED CIRCLE

As mentioned in Section 4.2 of the main paper, in this section we present quantitative results demonstrating the effectiveness of drawing red circles in reducing the sim2real gap. More specifically, we conduct an ablation using DynamiCrafter for the relative formulation of the elasticity property. The results are in Table 7. It can be observed that *red circle* can effectiveness mitigate the sim2real gap, as the performance on the real test split test-3 is significantly improved from $0.47$ to $0.84$.

Table 7: **Effectiveness of *red circle*** in reducing the sim2real gap. We compare the performance of DynamiCrafter for the relative formulation of the elasticity property, with and without the red circle drawn on the input video frames.

| Setting | Test-1 | Test-2 | Test-3 |
|---|---|---|---|
| With Red circle | 1.00 | 0.98 | 0.84 |
| Without red circle | 0.95 | 0.94 | 0.47 |

