# OpenReview forum: "Inferring Dynamic Physical Properties from Video Foundation Models"
_ICLR.cc/2026/Conference — ICLR 2026 Conference Withdrawn Submission_

### Official Review · Reviewer_XtX3 · 2025-10-18

**Soundness:** 2
**Presentation:** 2
**Contribution:** 2
**Rating:** 4
**Confidence:** 3

**Summary:**

This paper investigates the ability of modern video foundation models to infer dynamic physical properties from visual data. The authors focus on three properties that require temporal reasoning: elasticity, viscosity, and dynamic friction. To facilitate this study, they introduce a new benchmark dataset, PhysVid, which includes both synthetic videos with precise ground-truth values and real-world videos for evaluating generalization. The paper evaluates three distinct approaches: a classical computer vision "oracle" method designed to establish a performance upper bound, a simple readout mechanism on frozen features from a pre-trained Video Generative Model (VGM) and a Video Self-Supervised Model (VSM), and various prompting strategies for Multimodal Large Language Models (MLLMs). The experiments, conducted in both absolute value prediction and relative comparison settings, reveal that while VGMs and VSMs show promising capabilities, they still fall significantly short of the oracle, particularly on real-world data and for absolute value regression. The study finds that MLLMs currently perform poorly on this task, though their performance can be improved with more sophisticated prompting.

**Strengths:**

*Valuable New Benchmark (PhysVid)*: A major contribution of this work is the introduction of the PhysVid dataset. The combination of a large-scale synthetic dataset (with separate splits for in-distribution and out-of-distribution testing) and a real-world test set provides a valuable resource for the community. The careful construction to study properties like elasticity, viscosity, and friction is commendable.

*Comprehensive Evaluation Framework*: The experimental design is sound and insightful.
- The inclusion of an "oracle" model based on classical computer vision techniques provides a strong, interpretable upper bound on what is achievable from the visual data alone. This is crucial for contextualizing the performance of the foundation models.
- The comparison across three major classes of models—generative, self-supervised, and MLLMs—offers a broad snapshot of the current landscape and their respective strengths and weaknesses on this task.

**Weaknesses:**

*Limited Contribution Beyond Benchmarking*: While the evaluation is thorough, the paper's primary contribution is a benchmark and an analysis of existing models. It successfully highlights a performance gap but does not propose a novel method or training strategy to close that gap. For a top-tier conference like ICLR, a more substantial contribution would involve proposing a solution—such as a physics-informed pre-training objective, a specialized model architecture, or an improved fine-tuning technique—rather than solely documenting the limitations of current approaches. The work feels more like an excellent diagnostic study than a paper presenting a new advance.

*Narrow Scope of Evaluation*: The conclusions drawn are very broad ("Video Foundation Models"), but the evidence is based on a narrow selection of both physical properties and models.
- Limited Physical Properties: The study is confined to elasticity, viscosity, and dynamic friction. While these are good representatives, a more comprehensive benchmark would be necessary to make general claims about "physical reasoning." Other fundamental properties like mass, momentum, deformability, or fluid dynamics under different conditions are not explored.
- Limited Model Variety: The paper evaluates only one Video Generative Model (DynamiCrafter) and one Video Self-Supervised Model (V-JEPA-2). The findings cannot be confidently generalized to these entire classes of models. The video foundation model space is diverse and rapidly evolving. The inclusion of more, and potentially more recent, state-of-the-art models (e.g., Sora, Hunyuan-Video, or other leading VSMs) would be necessary to make the conclusions more robust and universally applicable.

*Significant Sim-to-Real Gap*: The models are trained exclusively on synthetic data, and the results reveal a significant performance drop on the real-world test-3 split, particularly for friction (as seen in Table 1). The authors acknowledge this and show that fine-tuning with a small real-world training set improves results (the * values in Table 1). However, this reliance on domain adaptation highlights a core weakness: the representations learned from simulation do not generalize well to the complexities of the real world. This casts doubt on the reliability of conclusions drawn primarily from synthetic data for real-world applications.

**Questions:**

- Given the observed performance gap between the foundation models and the oracle, what do the authors believe is the most promising path forward? Does the solution lie in better pre-training objectives, scaling up data, new model architectures, or more sophisticated fine-tuning methods?
- What was the reasoning for selecting these three specific physical properties? Are there plans to expand the PhysVid dataset in the future to cover a broader range of physical phenomena, which would allow for a more holistic evaluation of physical reasoning?
- The conclusions about the capabilities of VGMs and VSMs are based on a single model from each category. How confident are the authors that these results are representative of the entire model class? Would you expect a different SOTA model to perform substantially better or exhibit different failure modes?
- The paper highlights the sim-to-real gap, especially for friction. Does the necessity of fine-tuning on real data suggest that there are fundamental visual cues in real videos that are simply absent in the synthetic training data? If so, what might those be, and can simulators be improved to capture them?

**Details Of Ethics Concerns:**

The paper's use of real-world videos scraped from YouTube for the elasticity dataset (as mentioned on lines 154-155) raises significant ethical and legal concerns that are not addressed in the manuscript.
- Copyright and Licensing: The authors state on line 971 that "All datasets will be made publicly available upon paper acceptance under the CC-BY-4.0 license." This is a major concern. The authors do not own the copyright to the videos downloaded from YouTube. The content creators on YouTube license their work under specific terms (e.g., Standard YouTube License, or a Creative Commons license of their own choosing). The authors cannot unilaterally re-license this content as CC-BY-4.0. Distributing this data as proposed would likely constitute a copyright violation and a breach of YouTube's terms of service.
- Privacy: While the search term "ball bouncing experiments" seems benign, the collected videos could inadvertently contain personally identifiable information (PII), such as faces or voices of individuals in the background, or reveal private locations. The paper does not mention any steps taken to anonymize the data or to ensure that the privacy of the individuals in the videos is protected.
- Data Provenance and Consent: There is no discussion of the consent of the original video creators. While the videos are publicly accessible, this does not automatically grant permission for them to be included and redistributed in a machine learning dataset for research purposes.

---

> ### Author Response · Authors · 2025-12-01
>
> According to https://iclr.pangram.com/reviews?query=Inferring+Dynamic+Physical+Properties+from+Video+Foundation+Models&sort_by=submission_id_hash&sort_dir=asc&prediction_filter=&rating_filter=&confidence_filter=, this review seems to be fully AI generated.

---

### Official Review · Reviewer_m3k3 · 2025-10-31

**Soundness:** 3
**Presentation:** 3
**Contribution:** 3
**Rating:** 6
**Confidence:** 4

**Summary:**

This paper aims to evaluate the capabilities of current video foundation models in determining dynamical physical parameters from videos.
Three parameters are evaluated separately: elasticity, viscosity and dynamical friction.
Synthetic datasets are created with a simulator and real world test datasets are sourced online or recorded.
Three model classes are evaluated: generative diffusion, self-supervised, MLLM.
For the two former, extraction networks are trained and for the MLLMs, prompt strategies are crafted.
Performance is evaluated compared to engineered predictors based on visual cues.
It is found that models still strongly lack behind the engineered oracle in many settings with strong differences depending on model class and test set (synthetic/real).

**Strengths:**

- The paper tackles an interesting problem, extending previous benchmarks in a meaningful way. It provides a useful and challenging benchmark that could be used to grade improvements in video foundation models.
- The paper proposes evaluation strategies (e.g. prompting), useful for practitioners interested in similar tasks.
- Besides synthetic benchmarks, the paper also evaluates with real-world videos.
- The paper is mostly well written and clear.

**Weaknesses:**

Major:
- It seems that the absolute predictions differ strongly from ground truth values (appendix F, even for the oracle). If the prediction of absolute quantities is infeasible, this strongly questions the meaningfulness of the proposed quantities for benchmarking.
- Sec 4.2: Why L1 loss, not L2?
- The images and text labels in the figures are too small. They should be readable in printout (font size not smaller than 0.9 caption font size).
- l. 413, please specify the semantic cues.
- There is a large performance drop for all models including the video generative/self-supervised models on the test-3 split for elasticity absolute value estimation. This is not discussed in the text. Why does it happen?
- Augmenting of images with engineered heuristics (red circle) provides oracle information to all methods. The sim2real gap should be approached in a different manner, for example with more realistic simulation data.
- The paper should state if the proposed benchmark and datasets will be made publicly available to foster future research by the research community.

Minor:
- From the MLLM prompt extensions presented in sec 4.3, I found it hard to understand which variants are used in the experiments. Please explain clearly, which prompts are used in the experiments.
- It is hard to rate the performance of the models intuitively. Evaluation of human performance on the tasks could be insightful for comparison.
- Ablations in supplementary D and G are only evaluated on the elasticity task. How does it perform on the other tasks?

**Questions:**

- Regarding the major weakness, please elaborate. Why are predicted and ground truth values so different, even for the oracle?
  Why is the relative ranking important? Why are your proposed quantities still useful for benchmarking?

---

### Official Review · Reviewer_rtwz · 2025-10-31

**Soundness:** 3
**Presentation:** 3
**Contribution:** 3
**Rating:** 4
**Confidence:** 2

**Summary:**

This paper aims to examine whether current model can estimate physical properties from videos. To investigate this question, authors introduces PhysVid, a benchmark designed to evaluate understanding of dynamic physical properties such as elasticity, viscosity, and friction. A unified framework which includes oracle estimators, visual prompting for VFMs, and text prompting for MLLMs is proposed to test both quantitative and comparative reasoning. Experiments reveal that while VFMs capture partial physical cues, MLLMs still struggle with complex dynamics.

**Strengths:**

1. Benchmarking physical reasoning in current models is a meaningful and  essential question.
2. Well-constructed evaluation: Establish a set of estimators for elasticity, viscosity, and friction and include both synthetic and real test data.
3. Comprehensive results: Carefully design evaluation on different types of models with provide concrete analysis and direction for future research.

**Weaknesses:**

Major

1. Fair evaluation: Video Generative Model and Video SSL Model aggregates features using learnable queries while VLMs only use prompting technique and text as final answer, does it make a fair comparison across different models?
2. Following last question, It's interesting that VLMs lag behind visual models by large margin. As previous[1] work showed, text and visual information are not necessarily properly fused. I am curious whether this is same with physical properties? If similiar visual prompting (as in generative model and SSL model) are conducted on visual backbone of VLMs, what are results? Further, if the features are extracted from VLM hidden states, what are the results? Would be great if the authors can provide more analysis of where and why VLMs fail for inferring physical properties.
3. For Video Generative Model and Video SSL Model, learnable query and additional network may lead to some shortcut, especially the data used are relatively simple. I think a more direct observation maybe compare visual output, i.e. if given a short clip of a ball falling, can generation model generate a full sequence that match the oracle physical properties? From Tab.1. both Video Generative Model and Video SSL Model have relative reasonable abilities in understand physical dynamics, I wonder if this can be reflected in their output. Discussion on this problem would benefit this paper.

Minor

4. Data diversity: The data layout is relatively simple, which makes sense to focus only on physical properties. Apart from the synthetic-real distribution shift, I wonder if there is dynamics difference between train and test set? (e.g., test may complete different elasticity parameters?)



[1] Fu, Stephanie et al. 2025, Hidden in plain sight: VLMs overlook their visual representations

**Questions:**

See weakness

---

### Official Review · Reviewer_fJH5 · 2025-11-04

**Soundness:** 2
**Presentation:** 2
**Contribution:** 2
**Rating:** 2
**Confidence:** 3

**Summary:**

The paper considers the task of inferring physical properties from videos. Specifically, it considers three properties: elasticity, viscosity, and friction. To study this, the authors create a dataset of videos with corresponding labels. The dataset consists of videos from simulation and from the real world. The ground truth labels for simulation videos are obtained from the simulator and estimated in the case of real world videos. Given a video, the task is then to predict the label. The paper studies three different approaches that make use of classical computer vision, generative / self-supervised video models, and multimodal language models.

**Strengths:**

The problem of inferring physical properties from videos is interesting.

A dataset/benchmark to study it would be of the valuable to the community given the increasing interest in video models.

**Weaknesses:**

- I think it would be good to differentiate between kinematics and dynamics. The paper (and many other papers on video modeling) does not distinguish between these and often uses dynamics to mean kinematics. My understanding is that dynamics have to do with forces rather than observed changes in positions alone. It would be good to discuss this further and be more precise.
- I like the idea of being able to study this problem with a dataset and a benchmark (and particularly using simulated videos for which we can access ground truth values of dynamics properties) but I am not quite sure what the main takeaways are from the results. It would be good if the authors can comment on this.
- It would also be good to clarify what the primary contribution of the paper is. If it is studying this problem then the paper should discuss prior efforts and the relative tradeoffs. If the dataset is the primary contribution then the paper should discuss prior datasets used for studying this problem. Finally if it is about the findings then that should be highlighted. Currently it is a bit hard to understand the contributions and put them in context of prior work.

**Questions:**

Please see above.

---

### Note · Authors · 2025-12-21

I have read and agree with the venue's withdrawal policy on behalf of myself and my co-authors.